# microRNA-33 controls hunger signaling in hypothalamic AgRP neurons

Nathan L. Price[1,2,3,4,9], Pablo Fernández-Tussy [1,2,3,9], Luis Varela[2,3,5,6], Magdalena P. Cardelo [1,2,3], Marya Shanabrough [2], Binod Aryal[1,2,3], Rafael de Cabo [4], Yajaira Suárez [1,2,3,7], Tamas L. Horvath [2,3,5,6,8] ✉ & Carlos Fernández-Hernando [1,2,3,7] ✉

AgRP neurons drive hunger, and excessive nutrient intake is the primary driver of obesity and associated metabolic disorders. While many factors impacting central regulation of feeding behavior have been established, the role of microRNAs in this process is poorly understood. Utilizing unique mouse models, we demonstrate that miR-33 plays a critical role in the regulation of AgRP neurons, and that loss of miR-33 leads to increased feeding, obesity, and metabolic dysfunction in mice. These effects include the regulation of multiple miR-33 target genes involved in mitochondrial biogenesis and fatty acid metabolism. Our findings elucidate a key regulatory pathway regulated by a non-coding RNA that impacts hunger by controlling multiple bioenergetic processes associated with the activation of AgRP neurons, providing alternative therapeutic approaches to modulate feeding behavior and associated metabolic diseases.

The number of people in developed countries who are considered overweight or obese has increased dramatically over the last decade and is expected to continue rising. Overfeeding is one of the most important factors driving this obesity epidemic, which in turn promotes the development of cardiometabolic diseases like diabetes and atherosclerosis. Consistent with this, one of the most effective strategies for combating metabolic diseases is to enact lifestyle changes, including decreased food intake and increased exercise. However, the efficacy of dietary alterations is dependent upon compliance, which is difficult to maintain over long periods of time. As such, a more in depth understanding of the systems and pathways that regulate feeding behavior will provide important information for promoting and supporting healthy dietary practices.

microRNAs (miRNAs) and other non-coding RNAs have been demonstrated to be critical regulators of numerous physiologic conditions, and to play a key role in a multitude of different diseases, including those associated with metabolic syndrome[1,2]. Consistent with this, miRNAs have been shown to be necessary for proper maintenance of many essential metabolic functions, including modulation of feeding behavior by the hypothalamus[3]. The miR-33 family of miRNAs, consisting of miR-33a and miR-33b, are intronic miRNAs encoded within the sterol response binding protein 2 (*SREBP2*) and *SREBP1* genes, respectively[4,5]. miRNA-33a/b provide a critical link between the regulation of cholesterol and fatty acid biosynthesis by SREBPs, and cholesterol efflux, high-density lipoprotein (HDL) biogenesis, and fatty acid oxidation pathways[4–8]. Notably, pharmacological inhibition of miR-33 elevates hepatic ABCA1 expression, thereby increasing circulating HDL-C and attenuating the progression of atherosclerosis, highlighting the therapeutic potential of miR-33 inhibitors for the treatment of cardiovascular disease[9,10]. However,

[1]Vascular Biology and Therapeutics Program, Yale University School of Medicine, New Haven, CT, USA. [2]Department of Comparative Medicine, Yale University School of Medicine, New Haven, CT, USA. [3]Yale Center for Molecular and System Metabolism. Yale University School of Medicine, New Haven, CT, USA. [4]Experimental Gerontology Section, Translational Gerontology Branch, National Institute on Aging, National Institutes of Health, Baltimore, MD 21224, USA. [5]Laboratory of Glia -Neuron Interactions in the control of Hunger. Achucarro Basque Center for Neuroscience, 48940 Leioa, Vizcaya, Spain. [6]IKERBASQUE, Basque Foundation for Science, 48009 Bilbao, Vizcaya, Spain. [7]Department of Pathology. Yale University School of Medicine, New Haven, CT, USA. [8]Department of Neuroscience. Yale University School of Medicine, New Haven, CT, USA. [9]These authors contributed equally: Nathan L. Price, Pablo Fernández-Tussy. ✉e-mail: tamas.horvath@yale.edu; carlos.fernandez@yale.edu

work with genetic models of miR-33 deficiency has clearly demonstrated that global loss of miR-33 promotes the development of obesity and metabolic dysfunction[11,12]. These adverse metabolic effects were sufficient to offset the beneficial effects of miR-33 deficiency on macrophage cholesterol efflux, thereby negating any beneficial effects on atherosclerosis in the LDLR knockout model of atherosclerosis[13].

The obesity and metabolic dysfunction of miR-33 knockout mice were dependent upon increased food intake, but the mechansims by which miR-33 regulates feeding behavior remain unclear. As loss of miR-33 in peripheral organs that indirectly regulate feeding behavior, like the liver and adipose tissue, did not lead to increased body weight[14], we next sought to determine if miR-33 is directly involved in central regulation of feeding behavior. Within the arcuate nucleus of the hypothalamus, the agouti-related protein (AgRP)- and proopiomelanocortin (POMC)-expressing neurons are primarily responsible for promoting signals of hunger and satiety, respectively. Other neuronal cell types can also play a role in the regulation of feeding behavior, as well as astrocytes and microglia that can indirectly impact neuronal activity. Impairment in the activity of AgRP neurons has been shown to lead to a lean phenotype while activation of AgRP neurons increases feeding and promotes obesity[15–17]. In this work we demonstrate that selective loss of miR-33 in AgRP neurons is sufficient to increase feeding in HFD fed mice leading to the development of obesity and metabolic dysfunction.

## Results

### Genetic ablation of miR-33 in AgRP neurons causes metabolic dysfunction and obesity

As our previous work has demonstrated that loss of miR-33 in peripheral organs, such as the liver and adipose tissue, that indirectly regulate feeding behavior, did not lead to increased body weight[14], we next sought to determine if miR-33 is directly involved in regulating the activity of the AgRP and POMC neurons that promote signals of hunger and satiety. Our initial focus was the AgRP neurons based on earlier work connecting the activity of these hypothalamic hunger-promoting cells with intracellular lipid utilization[15] and the important role of miR-33 in regulation of lipid metabolism[4–8]. To determine the impact of removing miR-33 in AgRP neurons, we crossed our *miR-33^{flox/flox}* mice with both a conventional AgRP-Cre strain and a tamoxifen inducible *AgRP-ERT2^{Cre-Ai14}* strain. This allowed us to selectively remove miR-33 from AgRP neurons either from development on (*miR-33^{AgRPcKO}*) or selectively in adult mice (*miR-33^{AgRPiKO}*). We found that in both models, specific loss of miR-33 in AgRP neurons was sufficient to largely mimic the primary effects of global deficiency in response to HFD feeding. No apparent differences in body weight were observed between young *miR-33^{AgRPiKO}* mice and control animals when fed a chow diet. Upon HFD feeding, both male and female *miR-33^{AgRPiKO}* mice showed increased body weight and higher weight gain compared to control animals, which was maintained throughout the course of the experiment (Fig. 1a and c, Supplementary Fig. 1a, b). Similar effects were observed in *miR-33^{AgRPcKO}* mice (Supplementary Fig. 2a). Body composition analysis

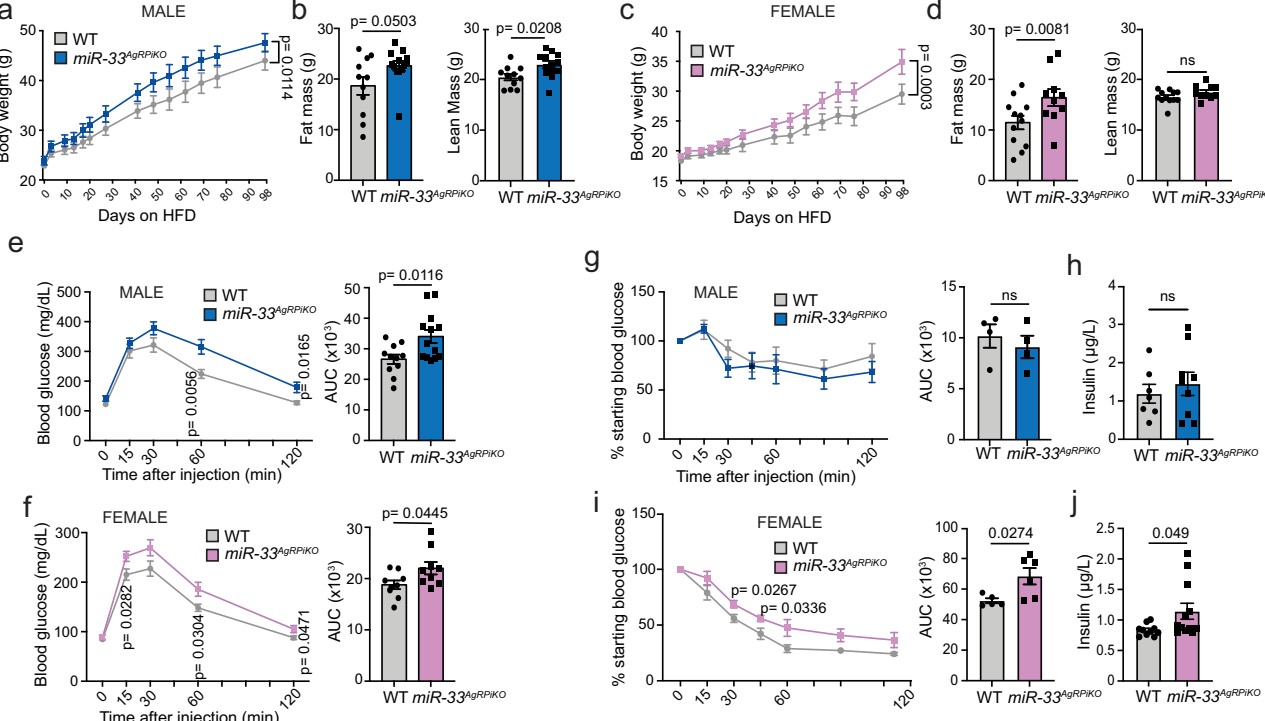

**Fig. 1 | Removal of miR-33 in AgRP neurons of adult mice promotes obesity and metabolic disfunction. a–d** Combined data on body weight (**a** and **c**), and fat and lean mass (**b** and **d**) from three independent cohorts of male (**a**, **b**) and female (**c**, **d**) wildtype (WT) and *miR-33^{AgRPiKO}* mice fed a high fat diet (HFD). **a** (*n* = 12 WT and 14 *miR-33^{AgRPiKO}* males); **b** (*n* = 11 WT and 13 *miR-33^{AgRPiKO}* males); **c** (*n* = 12 WT and 11 *miR-33^{AgRPiKO}* females); **d** (*n* = 12 WT and 10 *miR-33^{AgRPiKO}* females). **e**, **f** Changes in blood glucose levels during a glucose tolerance test (left panels) and the calculated area under the curve (right panels) from three independent cohorts of male (**e**) and female (**f**) WT and *miR-33^{AgRPiKO}* mice fed a HFD. **e** (*n* = 11 WT and 13 *miR-33^{AgRPiKO}* males); **f** (*n* = 9 WT and 9 *miR-33^{AgRPiKO}* females). **g** and **i**. Changes in blood glucose levels during an insulin tolerance test (left panels) and the calculated area under the curve (right panels) from one cohort of male (**g**) and two independent cohorts female (**i**) WT and *miR-33^{AgRPiKO}* mice fed a HFD. **g** (*n* = 4 WT and 4 *miR-33^{AgRPiKO}* males); **i** (*n* = 5 WT and 6 *miR-33^{AgRPiKO}* females). **h** and **j** Plasma insulin levels in three independent cohorts of male (**h**) and female (**j**) WT and *miR-33^{AgRPiKO}* mice fed a HFD. **h** (*n* = 7 WT and 9 *miR-33^{AgRPiKO}* males); **j** (*n* = 10 WT and 12 *miR-33^{AgRPiKO}* females). All data represent mean +/- SEM. Statistical Analysis assessed by unpaired 2-sided Student *t*-tests (**b**, **d**, **e**–**j**) or 2-way ANOVAs (**a** and **c**). Source data are provided as a Source Data file.

performed after 3 months of HFD feeding showed that these mice also had a significant increase in fat mass after HFD feeding (Fig. 1b and d, Supplementary Fig. 2b and e). In male *miR-33*$^{AgRPiKO}$ mice lean mass was also significantly increased, while female mice tended to have greater lean mass, but this was not significant. Consistent with these findings, the % fat mass was signficiantly increased while % lean mass was decreased in female *miR-33*$^{AgRPiKO}$ mice. No significant differences in % lean or fat mass were observed in males (Supplmentary Fig. 1c, d).

Similar to whole body miR-33 knockouts, both *miR-33*$^{AgRPiKO}$ and *miR-33*$^{AgRPcKO}$ also showed an impaired ability to regulate glucose homeostasis during a glucose tolerance test performed after 3 months of HFD feeding (Fig. 1e, f, Supplementary Fig. 2c and f). Female *miR-33*$^{AgRPiKO}$ also showed an impaired ability to reduce blood glucose in response to insulin (Fig. 1i) and higher fasting insulin levels (Fig. 1j), while male *miR-33*$^{AgRPiKO}$ mice did not show any indication of impaired insulin sensitivity (Fig. 1g, h). In order to more fully determine which of the metabolic deficiencies observed in global miR-33 knockout mice were apparent in our miR-33$^{AgRPiKO}$ mice, we have performed measurements of triglycerides and cholesterol in plasma and livers collected from these animals. These measurements demonstrate that miR-33$^{AgRPiKO}$ mice have increased triglyceridemia as shown by the higher levels of triglycerides in serum of both males and females after high fat diet feeding (Fig. 2a and d). On the other hand, serum cholesterol and HDL-cholesterol levels were slightly lower in males, while an increase in total cholesterol was observed in female mice (Fig. 2b, c, e and f). Additionally, miR-33$^{AgRPiKO}$ mice were found to accumulate greater amounts of triglycerides and cholesterol in the liver (including total cholesterol and free cholesterol) than WT mice upon high fat diet feeding. This was observed in both males and females (Fig. 2g–l).

As with whole body miR-33 knockout mice, the metabolic effects observed in *miR-33*$^{AgRPiKO}$ mice were likely due to changes in feeding behavior, as monitoring of individually housed mice demonstrated that *miR-33*$^{AgRPiKO}$ mice eat significantly more than controls when fed a HFD (Fig. 3a, b). While average daily food consumption was not found to be significantly different in chow diet fed mice with continuous

access to food (Fig. 3a, b), assessment of food consumption over a 24 h period following overnight fasting also revealed an increase in food consumption in both male and female *miR-33*$^{AgRPiKO}$ mice (Fig. 3c, d). Consistent with the notion that these effects may be due to differences in the activity of AgRP neurons, feeding in response to ghrelin was also enhanced in *miR-33*$^{AgRPiKO}$ mice (Fig. 3e). Circulating levels of ghrelin were not found to be different between high fat diet fed *miR-33*$^{AgRPiKO}$ mice and controls under fasted conditions (Supplementary Fig. 1e).

Because AgRP neurons have been shown to play an important role in central control of thermoregulation, we next sought to determine whether alterations in metabolic rate could contribute to the higher weight gain in *miR-33*$^{AgRPiKO}$ mice. Metabolic cage analysis performed in male mice after 1 month of HFD feeding did not reveal any differences between *miR-33*$^{AgRPiKO}$ and control mice in terms of the total levels of oxygen consumption, carbon dioxide production, or energy expenditure. Body mass has a major impact on respiratory parameters, and although not significantly different in the subset of animals used in this analysis, *miR-33*$^{AgRPiKO}$ mice had a trend toward higher average body weight. We therefore performed an analysis of covariance, which demonstrated that the relationship between these respiratory factors and body weight was significantly different, with *miR-33*$^{AgRPiKO}$ mice showing lower respiration than would have been predicted based on their weight (Fig. 4a–f, Supplementary Table 1). This could possibly have also contributed to the obesity phenotype observed in these animals. Additionally, the respiratory exchange ratio of *miR-33*$^{AgRPiKO}$ mice was lower, due to a higher reliance on dietary fats during the light cycle (Fig. 4g), while no differences were observed in activity (Fig. 4h). Together these findings demonstrate that the specific loss of miR-33 in AgRP neurons is sufficient to promote obesity and metabolic dysfunction in mice fed a HFD, similar to what was observed with global miR-33 deficiency.

## Loss of miR-33 in POMC neurons and astrocytes

POMC neurons play an important role in central regulation of feeding behavior. Thus, we also explored the impact of deleting miR-33 from these neurons by crossing our *miR-33*$^{flox/flox}$ mice with an established

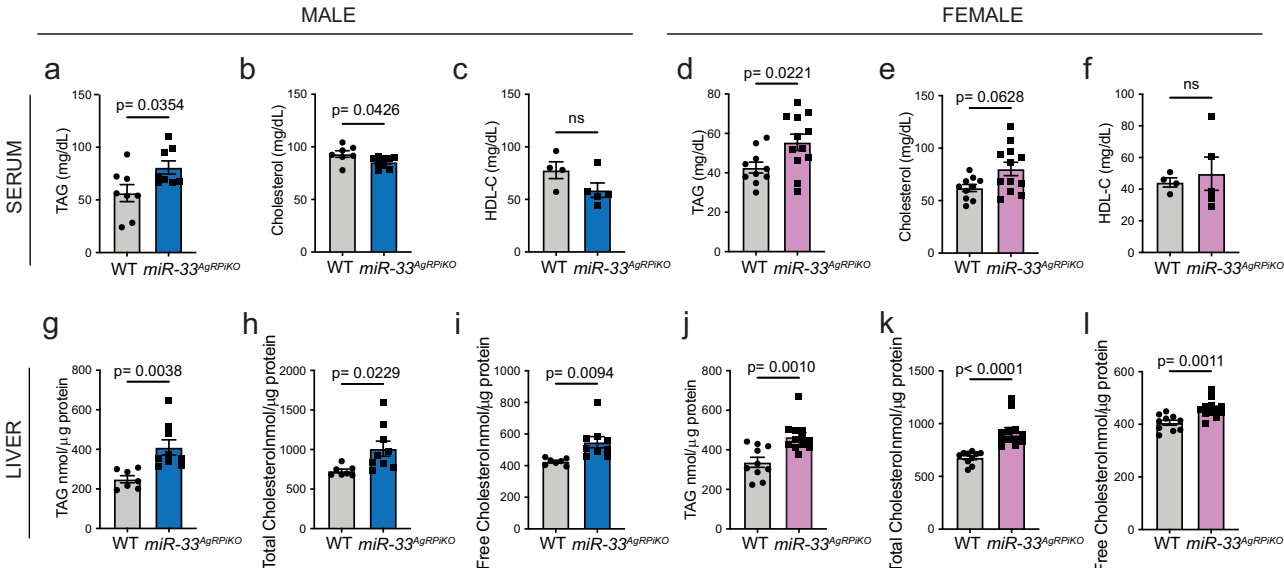

**Fig. 2 | Loss of miR-33 in AgRP neurons promotes lipid accumulation in circulation and in the liver. a–f** Circulating levels of tryglycerides (TAGs) (**a** and **d**), cholesterol (**b** and **e**), and high-density lipoprotein (HDL) cholesterol (C-HDL) (**c** and **f**) in plasma collected after an overnight fast in three independent cohorts of male (**a–c**) and female (**d–f**) wildtype (WT) and *miR-33*$^{AgRPiKO}$ mice fed a high fat diet (HFD). **a**, **b** (*n* = 7 WT and 9 *miR-33*$^{AgRPiKO}$ males; **c** *n* = 4 WT and 5 *miR-33*$^{AgRPiKO}$ males. **d**, **e** (*n* = 10 WT and 12 *miR-33*$^{AgRPiKO}$ females); **f** (*n* = 4 WT and 5 *miR-33*$^{AgRPiKO}$ females).

**g–l** Hepatic measurements of of tryglycerides (TAGs) (**g** and **j**), total cholesterol (**h** and **k**), and free cholesterol (**i** and **l**) in tissue collected after an overnight fast in three independent cohorts of male (**g–i**) and female (**j–l**) WT and *miR-33*$^{AgRPiKO}$ mice fed a HFD. **g–i** (*n* = 7 WT and 9 *miR-33*$^{AgRPiKO}$ males); **j–l** = 10 WT and 12 *miR-33*$^{AgRPiKO}$ females). All data represent mean +/- SEM. Statistical Analysis assessed by unpaired 2-sided Student *t*-tests (**a–l**). Source data are provided as a Source Data file.

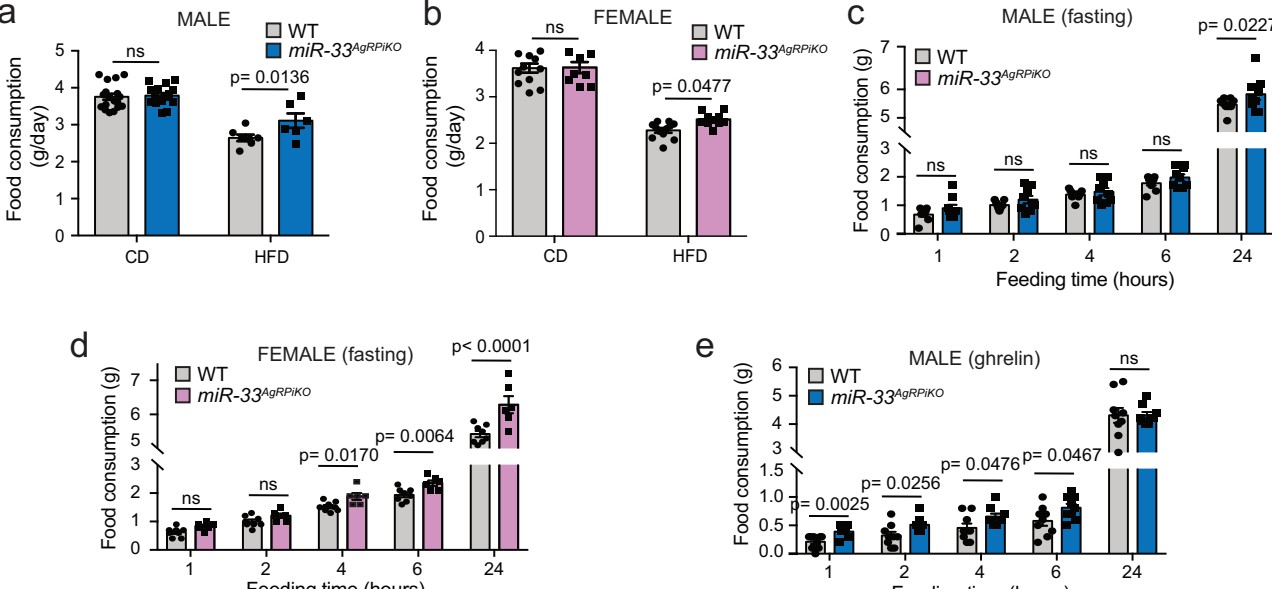

**Fig. 3 | Mice lacking miR-33 in AgRP neurons have increased food consumption.**
**a**, **b** Average daily food consumption in male (**a**) and female (**b**) wildtype (WT) and *miR-33^AgRPiKO* mice fed a chow diet (CD) (*n* = 18 WT and 16 *miR-33^AgRPiKO* males; *n* = 12 WT and 10 *miR-33^AgRPiKO* females) or high fat diet (HFD) (*n* = 7 WT and 6 *miR-33^AgRPiKO* males; *n* = 12 WT nd 9 *miR-33^AgRPiKO* females). **c**, **d** Direct measurements of food consumption over time in chow diet fed individually housed male (**c**) and female (**d**)

WT and *miR-33^AgRPiKO* mice fasted for 24 h (*n* = 9 WT and 9 *miR-33^AgRPiKO* males; *n* = 8 WT and 6 *miR-33^AgRPiKO* females). **e** Direct measurements of food consumption over time in individually housed chow diet fed WT and *miR-33^AgRPiKO* male mice treated with ghrelin (*n* = 9 WT and 9 *miR-33^AgRPiKO*). All data represent mean +/- SEM. Statistical Analysis assessed by unpaired 2-sided Student *t*-tests (**a**–**e**). Source data are provided as a Source Data file.

POMC-Cre conditional knockout model (*miR-33^POMCcKO*) (Supplementary Fig. 3a). We observed that male mice lacking miR-33 in POMC neurons did not show any differences in body weight or fat mass in response to HFD feeding compared to control animals (Supplementary Fig. 3b, c). Moreover, we did not observe any differences in feeding behavior (Supplementary Fig. 3d) or impairment in regulation of metabolic function (Supplementary Fig. 3e) in *miR-33^POMCcKO* mice.

Astrocytes have also been shown to control the activity of central feeding circuits and related behavioral and autonomic outputs[18–21]. Therefore, next we selectively deleted miR-33 from astroglia by crossing our *miR-33^flox/flox* mice with an inducible *GFAP-ERT2^Cre* model (*miR-33^GFAPiKO*) (Supplementary Fig. 3f). While the limited number of animals used in this assessment curtails our ability to draw firm conclusions, selective removal of miR-33 from astrocytes did not appear to result in any differences in body weight, fat mass, or regulation of glucose homeostasis (Supplementary Fig. 3g–i).

### miR-33 controls mitochondrial density and coverage in AgRP neurons

Overall, the phenotype of our *miR-33^AgRPKO* mice indicated that the ability of miR-33 to regulate feeding behavior may be due to direct effects of miR-33 on the activity and function of AgRP neurons. miR-33 has previously been shown to suppress numerous factors involved in mitochondrial biogenesis and function, including AMP-activated protein kinase subunit-a (*Prkaa1*), peroxisome proliferator-activated receptor gamma, coactivator 1 alpha (*Ppargc1a*), and carnitine palmitoyl-transferase 1 (*Cpt1a*)[6,22–25]. These same factors play critical roles in the activation of AgRP neurons[26,27]. To assess whether the increased feeding behavior in *miR-33^AgRPiKO* mice may be linked to alterations in mitochondrial abundance and function, as we reported previously[26,28,29], we performed electron microscopical analysis on AgRP neurons (Fig. 5a). This analysis demonstrated that the density and overall coverage of mitochondria was increased in AgRP neurons lacking miR-33 (Fig. 5b–e).

To further interrogate the mechanism by which loss of miR-33 may mediate the aforementioned effects, we crossed our *miR-33^AgRPiKO*

mice to the RiboTag mouse model (*B6J.129(Cg)-Rpl22^tm1.1Psam/SjJ*). These mice express an unmodified copy of the core ribosome subunit RPL22 under normal conditions, which is excised in cells expressing CRE, leading to the expression of a modified RPL22 construct containing multiple HA tags[30,31]. This allowed us to perform immunoprecipitation of HA-tagged ribosomes to assess the levels of ribosome associated mRNAs only from CRE⁺ AgRP neurons (Fig. 5f). Using this approach, we were able to demonstrate that AgRP neurons lacking miR-33 had significantly higher levels of actively translated miR-33 target genes, including multiple factors involved in mitochondrial function and biogenesis (Fig. 5g).

### Loss of miR-33 triggers AgRP neuronal activation

To provide further insight into the mechanisms by which miR-33 affects the function of AgRP neurons, we performed single cell RNA sequencing (scRNA-seq) on cells isolated from the hypothalamus of control and *miR-33^AgRPiKO* mice (Fig. 6a–c). This analysis demonstrated that AgRP neurons lacking miR-33 had significantly increased expression of AgRP (Fig. 6c). Numerous pathways related to neuronal function were also elevated in these cells (Fig. 6d, Supplementary Data 1). Consistent with the results from our RiboTag mouse model, the number of AgRP neurons expressing miR-33 target genes involved in mitochondrial function and biogenesis was significantly higher in the absence of miR-33. Within these cells, the expression levels were only found to be significantly altered for *Ppargc1a*, likely due to the low number of AgRP+ cells and limited depth of sequencing (Fig. 6e). This analysis also demonstrated that the number of AgRP neurons expressing *Fos* and *Npy*, were elevated in *miR-33^AgRPiKO* mice, as was the expression of *Npy* which is indicative of an increase in the activity of these neurons (Fig. 6f). This finding was further supported by direct staining for cFOS in hypothalamic sections of control and *miR-33^AgRPiKO* mice expressing the TdTomato reporter, which showed a significant increase in the number of cFOS positive AgRP neurons (Fig. 6g). Further analysis of our single cell sequencing data demonstrated that of the 2698 upregulated genes in AgRP neurons lacking miR-33, 147 genes

were also strong predicted targets of miR-33 based on the TargetScan7.2 database (Fig. 6h, Supplementary Data 2 and 3). Interestingly, some of these upregulated predicted targets are involved in functions related to the activation and signaling in AgRP neurons, including GABAergic signaling related genes[32,33], calcium transport and singaling[34], and other lipid metabolism related genes associated with cholesterol and ceramide metabolism[35], suggesting that additional miR-33 targets may also be involved in mediating some of these effects.

## Discusion

Previous work from our group and others has shown that despite the beneficial effects of miR-33 inhibition on reducing atherosclerotic plaque formation[9,10] and preventing fibrosis and organ disfunction in multiple tissues[36], genetic deletion of miR-33 leads to increased food consumption and the development of obesity and metabolic dysfunction[11,12]. These effects were not observed in mice where miR-33 was selectively removed from the liver, adipose tissue, or macrophages[14]. Other studies have shown that miR-33 in specific subsets of neurons can regulate state dependent memory and adaptive thermogenesis through repression of gamma-aminobutyric acid (GABA)$_A$ receptor subunits, but an impact on obesity was not observed[37,38]. We now demonstrate that miR-33 plays an important role

in control of hypothalamic AgRP neuronal activity. Specific removal of miR-33 from these cells resulted in a significant increase in multiple miR-33 target genes, including those involved in mitochondrial function and long chain fatty acid metabolism, resulting in increased activity of AgRP cells. These results indicate that by the combined regulation of these related factors, miR-33 plays a critical organizational role in suppressing the activity of AgRP neurons, thereby limiting feelings of hunger and reducing the development of obesity and metabolic dysfunction (Fig. 7). As miR-33 family members are encoded within the genes of the SREBP transcription factors, additional links between lipid metabolism and regulation of AgRP neurons should be a topic of future work. As a key regulator of AgRP neurons and the promotion of hunger signals, targeted approaches to enhance the function of miR-33 in AgRP neurons could provide a novel approach to modulate feeding and prevent or limit the development of numerous metabolic conditions associated with excessive nutrient intake. On the other hand, increased activity of these cells is associated with calorie restriction, the most robust intervention to promote health- and life-span across species[39]. Finally, AgRP neuronal activity was tied to control of the cerebral cortex and complex behaviors[40,41] and their disorders, including anorexia nervosa[42]. Thus, the involvement of miR-33 in AgRP neuron-related propagation of both negative and beneficial effects on health need to be pursued.

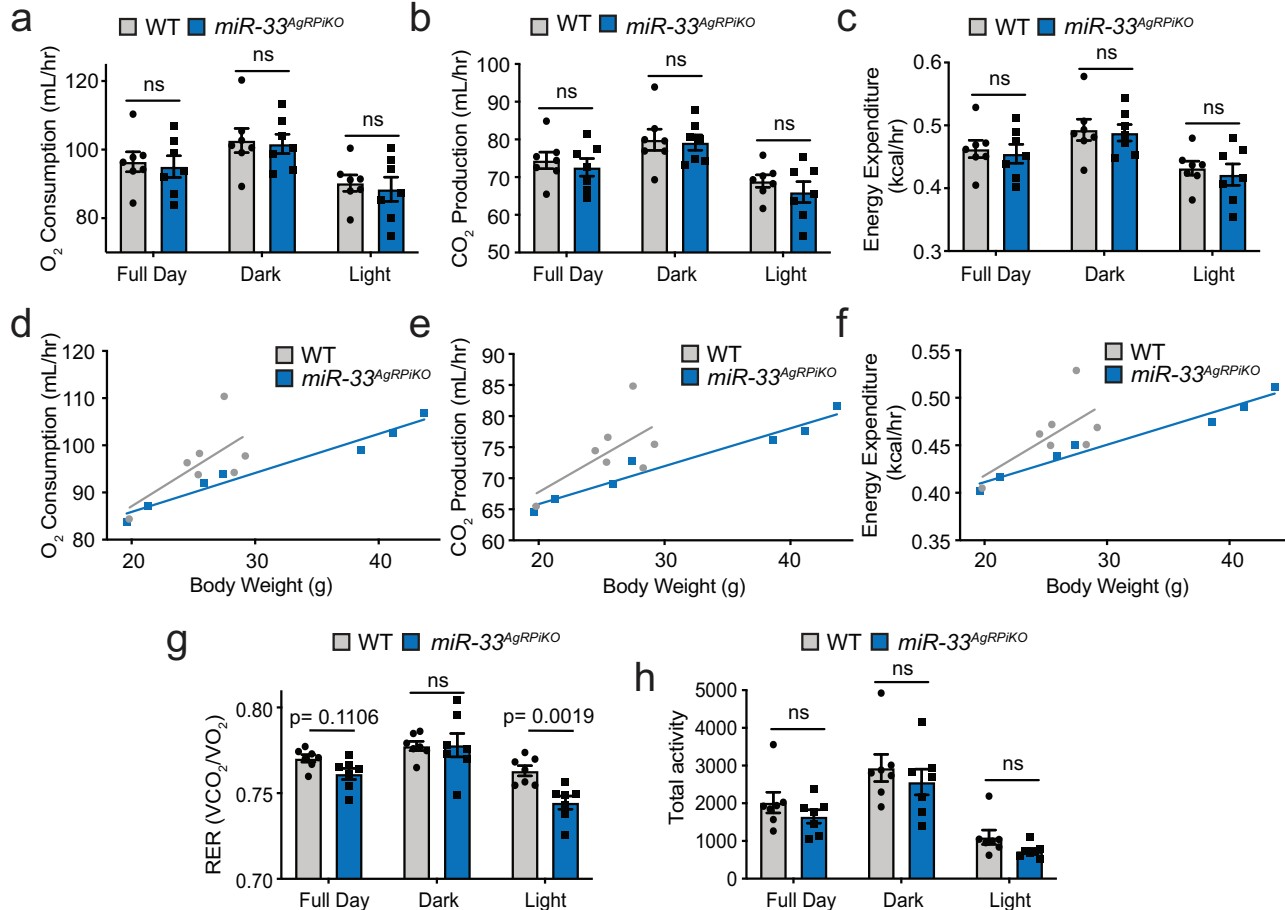

**Fig. 4 | Removal of miR-33 from AgRP neurons alters metabolic rate.** Metabolic cage analysis in a single cohort of male wildtype (WT) and *miR-33^{AgRPiKO}* mice after 1 month of high fat diet (HFD) feeding. **a–c** Average measurements of oxygen (O₂) consumption (**a**), carbon dioxide (CO₂) production (**b**), and energy expenditure (**c**) across the full day or 12 h dark and light cycles in WT and *miR-33^{AgRPiKO}* mice (n = 7). **d–f** Regression plots of average daily O₂ consumption (**d**), CO₂ production (**e**), and energy expenditure (**f**) as a function of body weight in WT and *miR-33^{AgRPiKO}* mice

(n = 7 WT and 7 *miR-33^{AgRPiKO}*). Statistical analysis was performed using GLM ANCOVA (Supplementary Table 1). **g, h** Average measurements of respiratory exchange ratio (RER) (**g**) and activity (**h**) across the full day or 12 h dark and light cycles in WT and *miR-33^{AgRPiKO}* mice (n = 7). All data represent mean +/- SEM. Statistical Analysis assessed by ANOVAs (**a–c, g–h**). Source data are provided as a Source Data file.

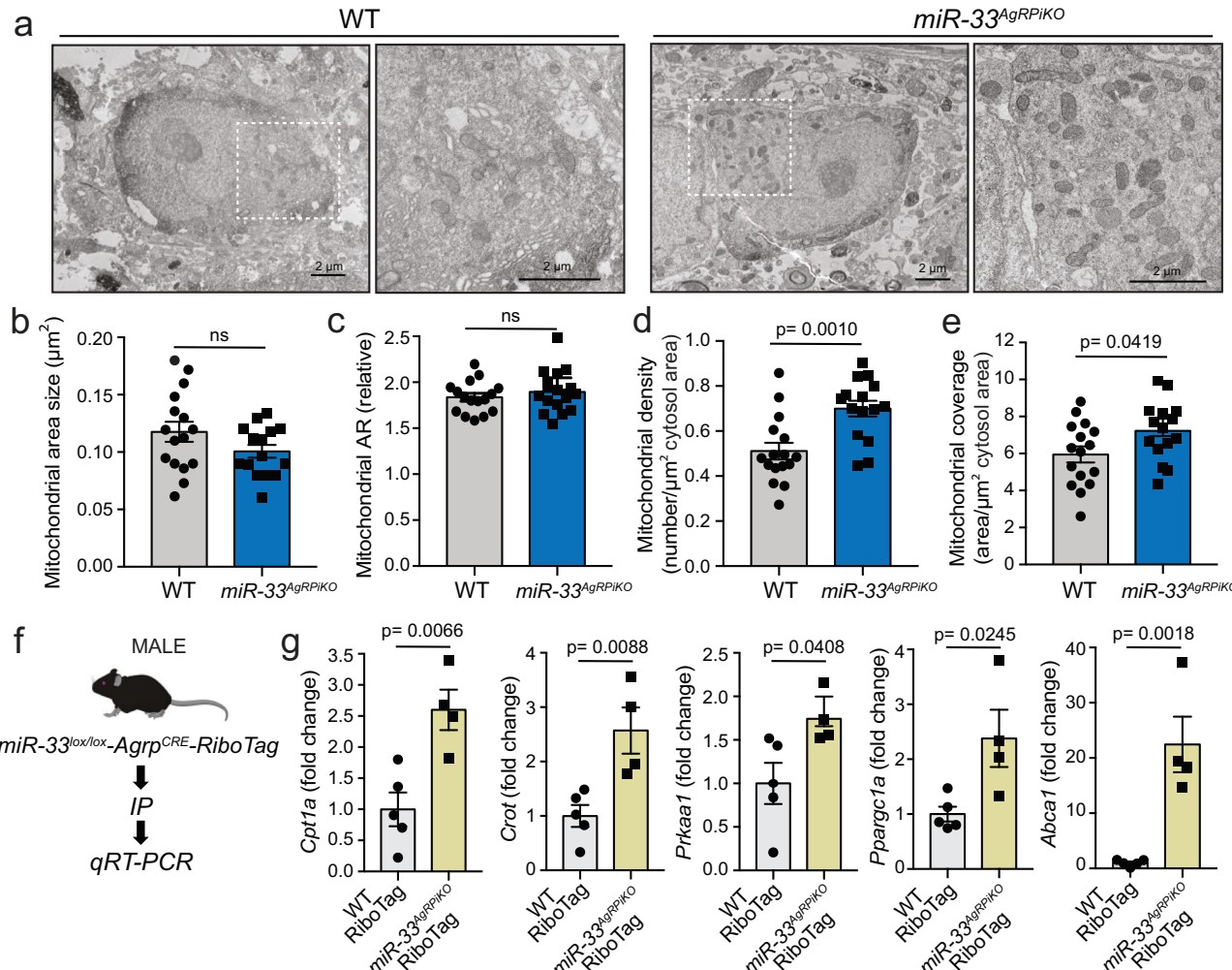

**Fig. 5 | AgRP neurons lacking miR-33 show increased mitochondrial density and elevated expression of miR-33 target genes. a** Representative images of electron microscopy (EM) imaging in sections of the arcuate nucleas from fed wildtype (WT) and *miR-33^AgRPiKO* mice after high fat diet (HFD) feeding. **b–e** Quantification of mitochondrial size (**b**), aspect ratio (**c**), density (**d**), and coverage (**e**) from analysis of EM images of WT and *miR-33^AgRPiKO* mice (*n* = 3 WT and 3 *miR-33^AgRPiKO*). **f** Schematic representation demonstrating pulldown of ribosome associated mRNAs from AgRP neurons using RiboTag mice. Generated using BioRender.com. **g** Relative expression of actively translated miR-33 target genes in WT and *miR-33^AgRPiKO* mice after overnight fasting (*n* = 5 WT and 4 *miR-33^AgRPiKO*, hypothalamus from 4 mice were pooled per "n" sample in order to get enough material for RNA processing). All data represent mean +/- SEM. Statistical analysis assessed by unpaired 2-sided Student *t*-tests (**b–g**). Source data are provided as a Source Data file.

miRNAs are important regulators of nearly all biologic functions and have been shown to contribute to the development of a broad array of disorders. Inhibition of miR-33 in particular is a promising therapeutic approach for many diseases. miR-33 was first characterized as a key regulator of cholesterol metabolism[4–8]. Work by different groups demonstrated that inhibition of miR-33 can protect against the development of atherosclerosis[9,10], which was later shown to be primarily due to promotion of reverse cholesterol transport in arterial macrophages by disruption of miR-33 mediated inhibition of ABCA1. This was accomplished through the creation of a binding site mutant mouse model in which the ability of miR-33 to bind mRNA of *Abca1* was specifically disrupted[43]. This study demonstrated that disrupting a single miRNA/target interaction was sufficient to impede the ability of a miRNA to mediate a complex physiologic function in vivo, as most miRNAs exert their effects by targeting multiple different mRNAs within the same or related pathways. More recently, work with tissue or cell type specific knockout models or targeted inhibition has shown that miR-33 promotes the development of fibrosis and inflammation in multiple different organs, including the kidney, liver, and lung[14,36,44]. These effects are believed to be mediated, at least in part, through repression of cellular engergetics, likely through direct targeting of

many different factors involved in mitochondrial biogenesis (*Prkaa1*, *Ppargc1a*), and fatty acid oxidation (*Cpt1a, Crot*).

Global deletion of miR-33 was found to promote obesity and metabolic dysfunction due to excess feeding[11,12]. Our previous work with liver and adipose specific miR-33 knockout strains had not shown a feeding/obesity phenotype[14]. Therefore, we wanted to determine whether miR-33 may play a role in mediating the activity of some of the primary neuronal drivers responsible for central regulation of feeding behavior. Our initial focus was the AgRP neurons based on our earlier work connecting the activity of these hypothalamic hunger-promoting cells with intracellular lipid utilization[26], and the important role of miR-33 in regulation of lipid metabolism[4–8]. Our work demonstrates that specific removal of miR-33 from AgRP neurons results in increased food consumption, body weight gain, and adiposity in animals fed a high fat diet, resulting in impaired regulation of glucose homeostasis. Although differences in body composition could be contributing to the effect observed in glucose response, this is unlikely to be the case as similar response was observed in males and females and no changes in the percent of body composition was observed in males. Similar to global miR-33 knockout mice, *miR-33^AgRPiKO* mice also had higher levels of circulating lipids and increased accumulation of triglycerides and

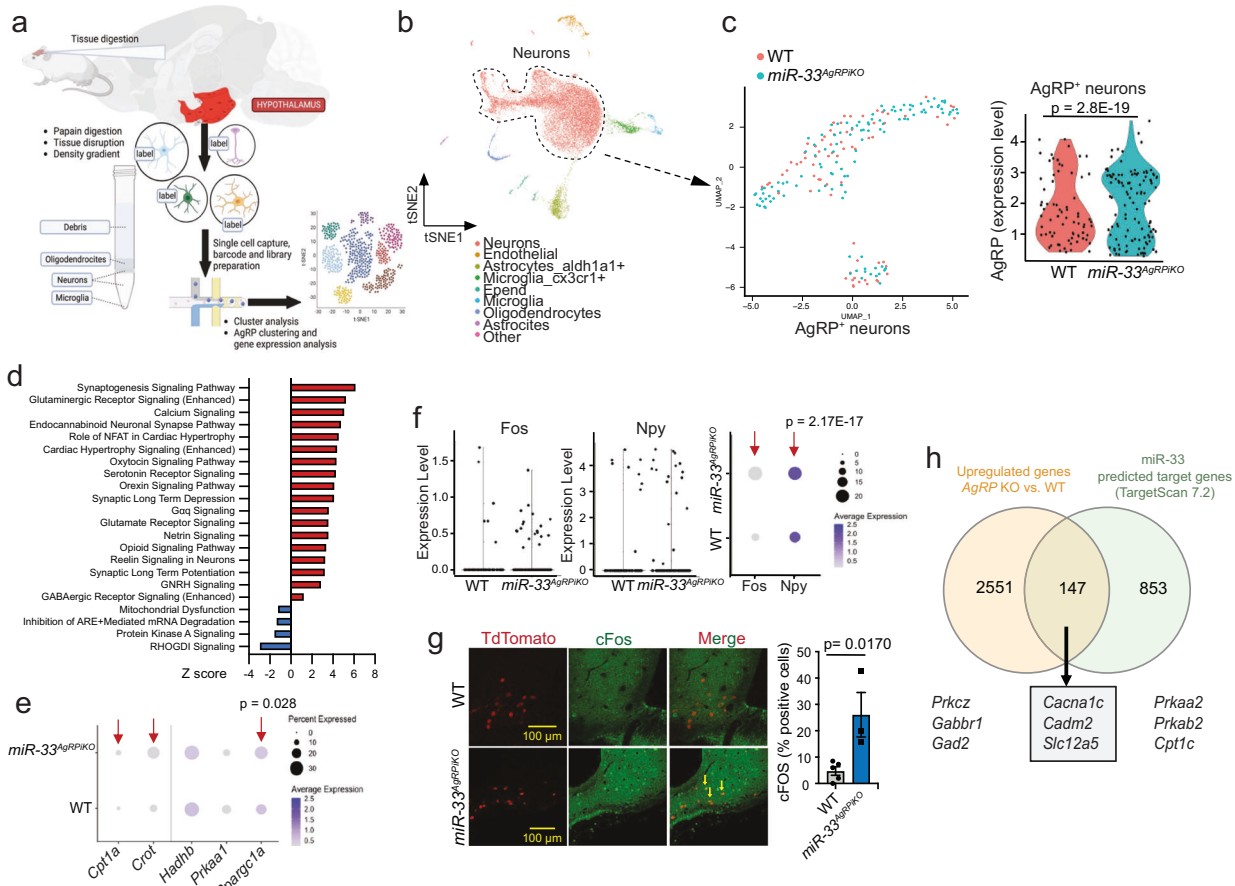

**Fig. 6 | Loss of miR-33 in AgRP neurons promotes changes associated with increased activity and mitochondrial biogenesis/function. a** Schematic depiction of approach used for performing scRNA-seq in cells isolated from the hypothalamus of wildtype (WT) and *miR-33^AgRPiKO* mice under fed condition after high fat diet (HFD) feeding. Generated using BioRender.com. **b** tSNE plot of cell clusters identified from SC-seq analysis. **c** UMAP projection of single cell profiles from WT (red) and *miR-33^AgRPiKO* (blue) mice of AgRP+ positive neurons identified from SC-seq analysis and violin plots of AgRP expression in AgRP+ neurons from WT and *miR-33^AgRPiKO* mice. **d** Canonical pathways represented by Z-score among differentially expressed genes in scRNA-seq analysis of AgRP neurons from WT and *miR-33^AgRPiKO* mice. Red bars indicate pathways in which genes are consistently upregulated, and blue downregulated based on the predicted *Z*-score. All represented pathways were significantly changed with a −Log *P*-value > 1.3. **e** Dot plots representing the

percentage of AgRP neurons expressing miR-33 target genes and relative expression within these cells. Red arrows indicate differences in the percentage of cells expressing a gene. **f** Violin plots and Dot plots representing the percentage of AgRP neurons expressing miR-33 target genes related to AgRP neuronal activity. Red arrows indicate differences in the percentage of cells expressing a gene. **g** Representative images and quantification of immunofluorescent staining of cFOS (green) in tdTomato + (red) AgRP neurons (*n* = 5 WT and 3 *miR-33^AgRPiKO*). Data represents mean +/- SEM. **p* < 0.05 as assessed by unpaired 2-sided Student *t*-test. **h** Venn diagram of overlap between upregulated genes in AgRP neurons lacking miR-33 and the top 1000 predicted miR-33 target genes from TargetScan7.2. Table including a list genes found to be relevant with AgRP functions. Source data are provided as a Source Data file. **a–g** (*n* = 8 WT and 6 *miR-33^AgRPiKO*).

cholesterol in the liver. While loss of miR-33 in AgRP neurons was sufficient to promote many of the phenotypes observed in global miR-33 knockout mice fed a high fat diet, the changes observed in these animals were generally more moderate that those observed in whole body knockouts. This suggests that other tissues or cell types may also play a role in mediating some of these phenotypes. Indeed, other work has recently shown that removal of miR-33 from dopamine-β-hydroxylase (DBH) positive cells reduced adaptive thermogenesis[37]. While these mice were not reported to have altered body weight, it is possible that reduced metabolic rate due to altered thermogegenesis may have played a role in helping to further promote the obesity related phenotypes observed in global miR-33 knockouts.

Consistent with the known role of miR-33 in regulating cellular bioenergetics, numerous miR-33 target genes involved in mitochondrial function and fatty acid metaboloism were found to be upregulated in AgRP neurons lacking miR-33. Cellular bioenergetics is known to impact the activity of AgRP and POMC neurons in the hypothalamus. Interestingly, many of the miR-33 targets involved in regulation of cellular bioenergetics have also been found to be important regulators

of AgRP neurons. Specific deletion of exons 3 – 5 of the *Ppargc1a* gene in AgRP neurons has been shown to reduce food intake and the induction of AgRP expression in response to fasting[27]. Similarly, hypothalamic AMPK has been shown to be important for the ability of the hypothalamus to properly regulate feeding behavior in response to changes in nutrient status, and inhibition of hypothalamic AMPK impairs the regulation of feeding behavior in response to fasting and refeeding or stimulation with hormones like ghrelin and leptin[26,45]. Consistent with this, deletion of AMPK in AgRP neurons resulted in reduced body weight over time[46]. Targeted deletion of CPT1 in AgRP neurons resulted in reduced body weight in both sexes, although feeding was only found to be reduced in male mice, while females demonstrated similar feeding but an increase in energy expenditure[47]. We have shown earlier that AMPK and its downstream targets, including UCP2, have critical roles in control of mitochondrial oxygen consumption and dynamics[26]. While these findings suggest that miR-33 may modulate the activity of AgRP neurons at least in part by targeting multiple different bioenergetic pathways that are required for full acitvation of these neurons, at present there is no ex vivo or in vivo

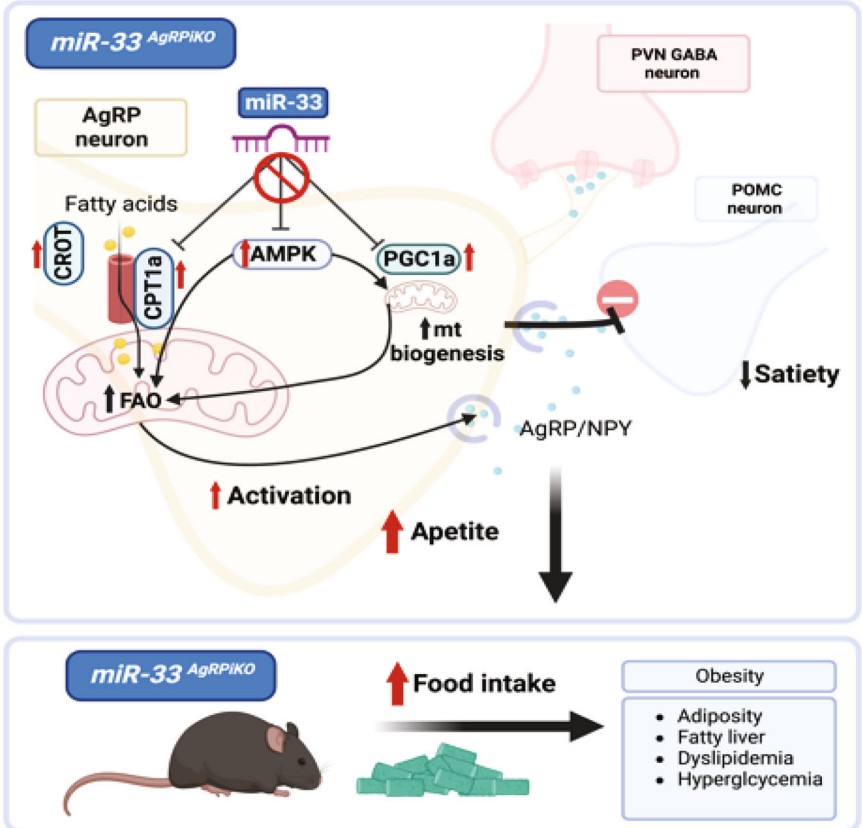

**Fig. 7 | Schematic representation of role of miR-33 in the regulation of AgRP neurons and the downstream impact on feeding and metabolism.** Generated with BioRender.com.

methodology to address these issues directly in AgRP neurons. As with many microRNAs, other factors are likely also involved in mediating these effects, since miR-33 is capable of targeting many different mRNAs, as suggested our scRNA-seq results. In the future, it will be important to study which of the predicted miR-33 target genes upregulated in the AgRP neurons lacking miR-33 are actually directly targeted by miR-33. Together these findings identify an unexpected level of regulation of feeding behavior by miR-33 and suggest a deeper link between regulation of lipid metabolism and control of feeding behavior.

**Study limitations**

While this work demonstrates that selective removal of miR-33 is sufficient to elevate AgRP neuronal activity, increase food consumption, and promote obesity and metabolic dysfunction in both male and female mice fed a high fat diet, this study did not determine whether the more modest difference in body weight and metabolic function that occurred with age in chow diet fed mice would also be observed in miR-33$^{AgRPiKO}$ mice. In the future it will be important to evaluate miR-33$^{AgRPiKO}$ mice over a longer period of time in the absence of HFD feeding to more fully determine the extent to which these animals mimic the phenotype of mice lacking miR-33 globally. Additionally, the measurments of respiratory parameters were only performed in male mice and the number of animals included in this analysis was more limited than that used in the overall assessment of weight gain and metabolic function. Body composition can have a major impact on these types of respiratory parameters, however body composition data was not collected at the time metabolic testing was performed, so determination of how normalization to fat or lean mass may impact these findings was not possible. Although our previous studies have demonstrated the efficacy and selectivity of the conditional miR-33

knockout allele in other tissues, and other works have also demonstrated the specificity of the AgRP-cre models, none of the approaches used in this work allowed us to directly asses miR-33 expression in these models. Finally, scRNA-seq was a valuable tool to evaluate the impact of miR-33 at the cellular level, providing insight into the genes and pathways disregulated in AgRP neurons lacking miR-33. However, the low number of AgRP neurons and the limited depth of coverage with single cell sequencing may be limiting in the statistical analysis in differential gene expression. At this time we are not able to directly determine which of the identified miR-33 targets are primarily responsible for mediating the altered feeding behavior of these mice, as this type of assessment would require the generation of multiple mouse models in which the ability of miR-33 to target individual mRNAs, and likely a combination of different mRNAs, is disrupted. To the best of our knowledge this has never been performed in a cell type specific manner as would be required to assess the specific effects in AgRP neurons.

## Materials and methods
### Animals
Generation of conditional miR-33 knockout mice *(miR-33$^{loxP/loxP}$)* was accomplished with the assistance of Cyagen Biosciences Inc. The success of this approach has been verified by Southern blotting and confirmed by PCR based genotyping using specific primers. To generate cell type specific miR-33 knockout mice, *miR-33$^{loxP/loxP}$* mice were crossed with AgRP-Cre or POMC-Cre strains to selectively remove miR-33 from AgRP neurons and POMC neurons, respectively. To remove miR-33 from astrocytes, *miR-33$^{loxP/loxP}$* mice were crossed to GFAP-ERT2$^{Cre}$ mice. Induction of CRE in this model was achieved by intraperitoneal injection of tamoxifen (100 mg/kg body weight) into *miR-33$^{flox/flox}$/GFAP-ERT2$^{Cre}$* and control mice for 5 consecutive days.

Experimental cohorts were produced by crossing *miR-33^{loxP/loxP}* males that were hetertozygous for Cre with *miR-33^{loxP/loxP}* females lacking Cre to generate control (*miR-33^{loxP/lox},Cre^-*) and cell type specific miR-33 knockout animals (*miR-33^{loxP/loxP},Cre^+*) These Cre-strains were kindly provided by the laboratory of Tamas Horvath. As leakiness has previously been reported for the constitutive AgRP-cre line, we prescreened mice generated with this model for non-specific miR-33 excision and limited our analysis to animals that did not show excision in other tissues.

To induce the removal of miR-33 from AgRP neurons of adult animals, we crossed our *miR-33^{loxP/loxP}* mice to a recently developed *AgRP-ERT2^{Cre-Ai14}* strain that was also provided by the Horvath lab. First, mice expressing a tamoxifen-inducible Cre recombinase (*CreERT2*) in cells expressing AgRP (*Agrp^{Cre:ERT2}*)[48] were crossed with Rosa26-lox-stop-lox-tdTomato (*Ai14*; cre-recombinase-dependent expression) mice (Ai14 reporter mice; stock #007914; The Jackson Laboratory, Bar Harbor, ME, USA) to label AgRP-expressing cells. *AgRP-ERT2^{Cre-Ai14}* mice have AgRP-expressing cells with the expression of tdTomato by tamoxifen administration. No observation of AgRP-tdTomato expression was found in the absence of tamoxifen administration, indicating that recombination was strictly dependent upon tamoxifen-induced Cre recombinase activation[49]. *miR-33^{loxP/loxP}* were crossed with *AgRP-ERT2^{Cre-Ai14}* mice to generated mice that were heterozygous for the miR-33 inducible knockout allele, with and without *AgRP-ERT2^{Cre-Ai14}*. These mice were then intercrossed to generate mice homozygous for the *miR-33^{loxP/loxP}* or WT allele with *AgRP-ERT2^{Cre-Ai14}*. Induction of CRE in this model was achieved by intraperitoneal injection of tamoxifen (100 mg/kg body weight) into *miR-33^{flox/flox}/AgRP-ERT2^{Cre-Ai14}* mice that have been fasted for 18 h to induce expression of AgRP. Six hours later, these mice are refed and allowed to recover for at least 48 h. This process is then repeated for a total of 5 times, resulting in efficient induction of CRE in AgRP neurons. *AgRP^{ERT2Cre-Ai14}* mice that do not have the floxed miR-33 allele serve as controls for these studies and undergo the same tamoxifen induction regime. the RiboTag mouse model [*B6J.129(Cg)-Rpl22^{tm1.1Psam}/SjJ*] was obtained from the Jackson Laboratory (Stock No. 029977) and crossed with *miR-33^{flox/flox}/AgRP-ERT2^{Cre-Ai14} or miR-33^{wt/wt}/AgRP-ERT2^{Cre-Ai14}* mice. All strains generated were backcrossed in to the *miR-33^{loxP/loxP}* to ensure the C57BL/6J genetic background before the experiments.

For diet induced obesity experiments, mice were fed a standard chow diet for 8–10 weeks after which were either switched to a high fat diet containing 60% calories from fat (D12492; Research Diets Incorporated, New Brunswick, NJ, USA) for 8–20 weeks or maintained on chow diet. Body weights were measured throughout HFD feeding studies and analysis of body composition was performed by Echo MRI (Echo Medical System). Mice used in all experiments were sex and age matched and kept in individually ventilated cages in a pathogen-free facility. All of the experiments were approved by the Institutional Animal Care Use Committee of Yale University School of Medicine.

### Glucose tolerance tests

Glucose tolerance tests (GTT) were performed following overnight fasting (16 h) by intraperitoneal (IP) injection of glucose at a dose of 1 g/kg body weight in mice fed a HFD for 3 months, following standard procedure[12]. Glucose was prepared at 10% (weight:volume) in PBS. Blood glucose was measured through tail vein incision using a Contour Ultra blood glucose meter at 0, 15, 30, 60, and 120 min post injection.

### Insulin tolerance tests

Insulin tolerance tests (ITT) were performed in mice fed a HFD for 3 months following a 6 h fasting by intraperitoneal (IP) injection of .75 u/kg insulin. Appropiate dose of insulin was diluted in PBS. Blood glucose measurements were taken through tail vein incision using a Contour Ultra blood glucose meter before and 15, 30, 45, 60, 90, and 120 min after injection of insulin.

### Measurement of circulating hormones

Circulating hormones were measured in mice serum by ELISA according to manufacturer's instructions: Insulin (Mercodia # 10-1247-01), Ghrelin (Sigma-Aldrich, #RAB0207). Serum was obtained from mice after fed a HFD for 3 months after overnight fasting.

### Measurement of circulating lipids

Mouse blood was obtained by cardiac puncture shortly after euthanizing the mice by isoflurane inhalation. Blood was immediately mixed with EDTA (0.5 M) (0.01:1, V:V) to avoid coagulation and kept at 4 °C until next step. Blood was then centrifuged, 7000 rpm, 8 min, 4 °C to obtain serum. HDL-C was isolated by precipitation of non-HDL-C with HDL precipitation reagent (Wako Pure Chemicals Tokyo, Japan, #997-01301). Briefly, plasma and HDL preicipating reagent were mixed at a ratio 1:1 and incubated at room temperature for 10 min. Then, mix was centrifuged 15 min, 3000 rpm, room temperature to precipitate non-HDL. Supernatant (HDL) was transferred to a new eppendorf and both HDL-C fractions and total plasma were stored at -80 °C. Total plasma cholesterol and TAGs were enzymatically measured (Wako Pure Chemicals Tokyo, Japan) according to the manufacturer's instructions.

### Measurements of liver lipids

Frozen livers were lysed in standard lysis buffer without detergent (50 mg/500μL). Following protein quantification, 400 μg/sample were brought to a total volume of 500 μL of NaCl (0.9%). Then, liver lipids were extracted using a solvent chloroform/methanol (2:1). 2 mL of the solvent were added to 500 μL of sample. Samples were then shaked 2 h, 1500 rpm and centrifuged for 10 min at 1500 rpm (4 °C). Lower phase containing lipids were taken and evaporated under a N2 flow. Dried pellets were resuspended in isopropanol (500μL) and TAG level in the liver was determined by using a commercially available assay kit (Sekisui Chemical Co.) according to the manufacturer's instructions. Total cholesterol and free cholesterol were measured with the AmplexRed commercial kit (Invitrogen), following manufacturer's recommendations.

### Feeding behavior

All measurements of feeding were performed in individually housed mice, as we described before[12]. Following at least 1 week of acclimation to individual housing, food consumption was monitored every 1–2 days and averaged over a 10 day period. For assessment of feeding in response to fasting, animals were fasted for 24 h beginning at 9:00 am. Food consumption was assessed 1, 2, 4, 6, and 24 h after feeding. For assessment of feeding in response to ghrelin, animals were injected with 200 nmol in 200 ml Ghrelin at 9:00am following overnight feeding. Food consumption was assessed 1, 2, 4, 6, and 24 h after injection.

### Electron microscopy

Mice (at least 3 per group) were euthanized by isoflurane inhalation and transcardially perfused with freshly prepared 4% PFA and 0.1% glutaraldehyde, as previously reported[19,21]. After post-fixation overnight, vibratome sections (50 μm) containing the ARC were immunostained with primary antibody anti-RFP (dilution 1:500, Rockland). After overnight incubation at room temperature, sections were washed with PB, incubated with biotin-conjugated donkey anti-chicken IgG (dilution 1:250, Jackson ImmunoResearch Laboratories) for 2 h, washed again, put in avidin–biotin complex (ABC; Vector Laboratories), and developed with 3,3-diaminobenzidine (DAB). Sections were then osmicated (15 min in 1% osmium tetroxide) and dehydrated in increasing ethanol concentrations. During the dehydration, 1% uranyl acetate was added to the 70% ethanol to enhance ultrastructural membrane contrast. Flat embedding in Durcupan followed dehydration. Ultrathin sections were cut on a Leica ultra-microtome, collected on Formvar-coated single-slot grids, and

analyzed with a Tecnai 12 Biotwin electron microscope (FEI) with an AMT XR-16 camera.

Hypothalamic sections containing AgRP immunoreactive cells (marked with TdTomato) with a visible nucleus were analyzed by electron microscopy. Mitochondrial cross-sectional area, and aspect ratio were calculated. Mitochondrial density was estimated by dividing the number of mitochondrial profiles by the cytosolic or cellular areas. Mitochondrial coverage was estimated by dividing the total area of mitochondria by the cellular area. Differences in the different mitochondrial parameters were tested using unpaired 2-sided Student $t$-test. $p \le 0.05$ was considered statistically significant.

### Indirect calorimetry (metabolic cages)
Indirect calorimetry was performed using an open-circuit, indirect calorimetry system (PhenoMaster, TSE systems) as previously described[20]. In brief, mice were trained for 3 days before data acquisition to adapt to the food/drink dispenser of the PhenoMaster system. Afterwards mice were placed in regular type II cages with sealed lids at room temperature (22 °C) and allowed to adapt to the chambers for at least 48 h. Food and water were provided ad libitum. All parameters were measured continuously and simultaneously. Statistical analysis was done using CalR: A Web-based Analysis Tool for Indirect Calorimetry Experiments.

### Immunofluorescence (cFOS)
WT and *miR-33*$^{AgRPiKO}$ mice were euthanized by isoflurane inhalation and transcardially perfused with 0.9% saline with heparin following 4% PFA. Post-fixed sections from WT and *miR-33*$^{AgRPiKO}$ mice were cut into 50 μm-thick sections. After 15 min washing in PB, the sections were incubated in blocking solution (1:20 normal donkey serum in PBS), containing 0.2% Triton X-100 for 30 min at room temperature. Sections were incubated with rb-anti-cFOS (1:2000, #sc-52-G, Santa Cruz) overnight at room temperature. The next day, sections were washed three times (5 min) in PB and incubated with the secondary antibody (donkey-anti-rabbit IgG fluor 488, 1:500, A-21206, Life Technologies) for 1 h at room temperature. The sections were coverslipped and imaged using a Keyence BZ-X700 fluoresce microscope[20,21].

### Ribosomal profiling
Translating ribosome affinity purification (TRAP) strategy was conducted in homogenate samples of mediobasal hypothalamus obtained from WT and *miR-33*$^{AgRPiKO}$ bred to RiboTag (*B6J.129(Cg)-Rpl22*$^{tm1.1Psam}$*/SjJ*) mice expressing Rpl22 and HA proteins in the ribosomes of AgRP neurons. Thereby, TRAP strategy allows for the immunoprecipitation of polysomes directly from AgRP-positive cells. Ribosome capture was performed as originally described with minor modifications[30,31]. Briefly, 1 month after last tamoxifen injection, 3-month-old mice (both male and female) were sacrificed and 4 hypothalami were pooled for each N. Total N used for the study was 4–5 per group. Tissues were homogenized in a 2 ml ice-cold Dounce homogenizer with 1 ml of cold homogenization buffer (HB-S: 50 mM Tris, pH 7.4, 100 mM KCl, 12 mM MgCl$_2$ and 1% NP-40 supplemented with 1 mM DTT, 1 mg/ml heparin, 100 μg/μl cycloheximide, 200 U/ml RNasin Ribonuclease inhibitor, and protease inhibitor cocktail). Lysates were centrifuged and a small aliquot from the lysates (20-25 μl) were kept for following Input analysis. Immunoprecipitation was performed by incubation of the sample with the anti-hemagglutinin (HA) antibody (1:400), 2 h, 4 °C. Sample with the antibody were incubated with 200 μl of G magnetic protein beads (Thermo Fisher), 4 h, 4 °C. Samples were then washed with high-salt buffer (50 mM TrisCl, pH 7.4, 300 mM KCl, 12 mM MgCl$_2$, 1% NP-40, 0.5 mM DTT, 100 μg/ml cycloheximide) and finally resuspended with RLT-β-mercaptoethanol buffer, following Qiagen RNeasy extraction kit protocol. RNA was isolated following Qiagen's RNeasy extraction kit directions.

Enrichment was confirmed by *Agrp* expression in immunoprecipitated samples compared to whole cell RNA input. After RNA isolation, standard rt-qPCR analysis was performed for the genes of interest. *Cpt1a* Mm (FW TTGATCAAGAAGTGCCGGACGAGT, R GTCCATCATGGCCAGCACAAAGTT), *Crot* Mm (FW ACTGA-GAGTGAAGGGCATTGTCCA, R AATGCCGCTATACTGGGTCCAACA), *Prkaa1* Mm (FW GAAAGTGAAGGTGGGCAAGC, R AAGGCTCC-GAATCTTCTGCC), *Ppargc1a* Mm (FW AACCCCAAGCGTCCGGCATG, R TGCGCTTTCTCAGGGTGGCG), *Abca1* Mm (FW AAAACCGCAGA-CATCCTTCAG, R CATACCGAAACTCGTTCACCC). Calculations were performed by a comparative method (2 − ΔΔCT) and relative expression was adjusted to 18 S (FW TTCCGATAACGAACGAGACTCT, R TGGCTGAACGCCACTTGTC) standard housekeeping gene. Quantitative PCR was performed on an CFX96 (BioRad). Gene expression differences were tested using unpaired 2-sided Student $t$-test. $p \le 0.05$ was considered statistically significant.

### Sample preparation for single cell RNA sequencing
Single cell suspensions were obtained from a pool of 6–8 freshly dissected hypothalamus. Tissue digestion was performed as previously described[50,51]. Briefly, 2 weeks after last tamoxifen injection, 3 month-old WT and *miR-33*$^{AgRPiKO}$ (both male and female) were fed a HFD for 1 month. Then, mice were anesthetized with isofluorane in non-fasted condition. Hypothalami were rapidly dissected and isolated from mice and place into a cold dish with 2 ml HABG buffer. Hypothalami were sliced into small pieces and then transferred to 50 ml tubes with 5 ml of HABG buffer. Then, hypothalami were incubated in water bath with smooth shaking at 30 °C, 8 min. Tissue was transferred to a new tube with 15 ml of HABG supplemented with papain (34 U/ml) (Worthington) and incubated at 30 °C, 30 min, with gentle shaking (170 r.p.m.) for enzymatic digestion. Then transfer tissue to a new 15 ml tube with 2 mL of HABG buffer and let 5 min at room temperature. Then, we proceeded to mechanical tissue disruption as follows. Hypothalami were triturated with a 9-inches Pasteur pipette with the fire-polished tip (10 times, 45 s, without introducing air bubbles). Tissue pieces were allowed to settle for 1 min and supernatant was transferred to a clean 15-ml tube. Tissue pieces were resuspended in 2 ml HABG, and trituration process was repeated two more times. In parallel, an OptiPrep 1.32 density gradient (Sigma-Aldrich) was prepared as follows: Layer 1 (173 μl OptiPrep + 827 μl medium); Layer 2 (124 μl OptiPrep + 876 μl medium); Layer 3 (99 μl OptiPrep + 901 μl medium); Layer 4 (74 μl OptiPrep + 926 μl medium). Cell suspension from supernatant mixes were carefully placed on top of the OptiPrep density gradient and centrifuged at 8 g, 15 min, 22 °C, no brake. Top 6 ml containing cell debris were removed and cell fractions 1–4 containing oligodendrocytes, neurons, and microglia were collected. Cell fractions were diluted with 5 ml HABG and centrifuged for 2 min at 200 g. Cell pellet was resuspended with 1 ml of HABG and filtered through a 40 μm cell strainer. Cells were counted and diluted at 1000 cells/μl and immediately processed for scRNA-seq.

### Single cell RNA sequencing procedure
Single cells obtained as above indicated, were encapsulated into droplets and processed following standard manufacturer's specification, using 10 X Genomics GemCode technology. Cells were loaded on a 10 x Genomics Chromium controller instrument to generate single-cell Gel Beads in emulsion (GEMS) at the Yale Center for Genome Analysis. Lysis and barcoded reverse transcription of polyadenylated mRNA from single cells were performed inside each GEM followed by complementary DNA (cDNA) generation using the single-Cell 3' Regent Kits version 2 (10 X Genomics). Libraries were sequenced on an Illumina HiSeq 4000 as 2 × 100 paired-end reads. Sc-RNA-sequencing data have been deposited in the Gene Expression Omnibus database (GSE222191).

## Single cell RNA-seq data analysis

Sample demultiplexing, aligning reads to the mouse genome (mm10 reference genome, University of California, Santa Cruz) with Software Tools for Academics and Researchers (STAR) and unique molecular identifier (UMI) processing. Data sets were processed using CellRanger software (version 2.1.1) as previously described. We identified 10000 WT cells (a mean of 27,361 reads per cell, and a median 282 genes per cell) and 10000 *miR-33*<sup>AgRPiKO</sup> cells (a mean of 23,718 reads per cell, and a median 389 genes per cell) for downstream analysis. Low-quality cells, doublets, and potentially dead cells were filtered based on the percentage of mitochondrial genes (40%) and number of genes and UMIs expressed in each cell.

Data clustering was performed using Seurat R package (version 4.1.0) in R (version 4.0.5) with filtered genes by barcode expression matrices as inputs (satija). Highly variable genes (HVGs) were calculated using Seurat function *FindVariableGenes* and used for downstream clustering analysis. Principal component analysis (PCA) was performed with *RunPCA* function (Seurat) using HVGs for dimensionality reduction and the number of significant principal components (PCs) was calculated using *JackStraw* function. We applied the *RunTSNE* function to significant principal components (PCs) and presented data in two-dimensional coordinates through t-distributed stochastic neighbor embedding generated by R package ggplot2. Alternatively, we also applied *RunUMAP* function to significant PCs identified by *JackStraw*. Clustering was done through *FindClusters* function using 30 significant PCs with a resolution of 0.2. Significantly differentially expressed genes in a cluster were analyzed using Seurat function *FindAllMarkers*, which were expressed in >25% of cells with at least 0.25-fold difference and reach statistical significance of an adjusted $P < 0.05$ as determined by the Wilcox test. For each cell type, we used multiple cell-type-specific/enriched marker genes that have been previously described to determine cell-type identity[52]. Specifically, we used *Ugt8a* and *Mbp* for oligodendrocytes, *Ndrg4* and *Stmn2* for neurons, *Fabp7* and *Ntsr2* for astrocytes, *Enkur* and *Foxj1* for ependymal cells, *Aif1* and *Cx3cr1* for microglia, *Slco1a4* and *Fn1* for brain endothelial cells. We then arranged all the identified cell types based on their expression profile into 6 classes of cell populations. From the 10000 cell from each genotype, we identified 7891 WT and 8160 *miR-33*<sup>AgRPiKO</sup> neurons. Then, 73 WT and 116 *miR-33*<sup>AgRPiKO</sup> AgRP positive neurons were selected for further analysis. To identify differentially expressed genes in AgRP neurons, Seurat *FindAllMarkers* function with *Poisson* test for UMI-based dataset was used, a $P < 0.05$ was considered significant.

Ingenuity Pathway Analysis (Ingenuity Systems QIAGEN, content version: 47547484, 2019) was used to determine pathways with differentially expressed genes across AgRP neurons. Only genes with an average $\log2FC > 0.25$ and a $P < 0.05$ were included for Ingenuity Pathway Analysis studies.

**Statistical analysis.** The number of animals used in each study is listed in the figure legends. Data are expressed as average ±SEM. Statistical differences were measured using unpaired 2-sided Student t-tests and 2-way ANOVAs with Bonferroni correction for multiple comparisons. Normality was checked using the Kolmogorov-Smirnov test. A value of $P \leq 0.05$ was considered statistically significant. Data analysis was performed using GraphPad Prism Software Version 7 (GraphPad). Metabolic cage data was analyzed using the CalRapp program. Differences in the relationship between respiratory parameters and body weight were analyzed using general linear model analysis of covariance (GLM ANCOVA), while RER and activity were analyzed by ANOVA. Specific statistics used for scRNA-seq and downstream analysis is detailed in Single-Cell RNA-seq data analysis section.

## Reporting summary

Further information on research design is available in the Nature Portfolio Reporting Summary linked to this article.

## Data availability

All data necessary to interpret the findings about the study are included within the manuscript or from corresponding author upon request. Sc-RNA-sequencing data have been deposited in the Gene Expression Omnibus database (GSE222191). Source data are provided with this paper.

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

## Acknowledgements

This work was at least in part supported by grants from the National Institutes of Health (R35HL135820 to CF-H; and R35HL155988 to YS; 1K01DK120794 to NP; and DK120891, DA046160, DK045735, AG067329 and DK126447 to TLH), the American Heart Association (20TPA35490416 to CF-H; and 874771 and 23CDA1055007 to PF-T), the Klarman Family Foundation to TLH and Plan Generación Conocimiento from the Spanish Ministry of Science and Innovation and Agencia Estatal de Investigación (PID2021-125193OA-I00 to LV). Schemes in Figs. 5f, 6a and 7 and Supplementary Figs. 2a and d and 3a and f were generated using BioRender.com.

## Author contributions

N.L.P, P.F.-T., T.H. and C.F.-H. designed research; N.L.P, P.F.-T., L.V., M.P.C, M.S. and B.A. performed research; N.L.P., P.F.-T., M.P.C and C.F.-H., N.L.P, P.F.-T., L.V., M.P.C, M.S. and B.A., R.C., Y.S., T.H. and C.F.-H. provided critical revision and discussion; and N.L.P, P.F.-T. and C.F.-H wrote the paper.

## Competing interests

The authors declare no competing interests.
