## [Peer Review File · Nature Communications]

microRNA-33 controls hunger signaling in hypothalamic AgRP neuronsREVIEWER COMMENTS

Reviewer #1 (Remarks to the Author):

The scope of the manuscript by Price et al. is likely appropriate for Nature Communications. It is a brief, but well-written, report on the role of miR-33 as a regulator of food-seeking behavior. Over-nutrition is a key component of the obesity epidemic that puts increasing strain on public health. Reversing obesity requires lifestyle interventions to include exercise and caloric restriction. However, the molecular mechanisms that regulate hunger are not fully defined. The present study adds detail to this knowledge gap by reporting that mice lacking miR-33 expression specifically in AgRP+ neurons have increased food consumption and obesity that is perhaps linked to enhanced mitochondrial biogenesis and function in AgRP+ neurons. The latter is supported by innovative and independent models of transcriptional profiling of AgRP+ neurons.

Conceptually, the story is interesting and stimulates new questions on the molecular mechanisms that regulate hunger and over-nutrition. The introduction is succinct and mostly straight-forward, but an improved transition between the last sentence to the results may help direct the reader. The discussion is also concise and direct. Although the results look solid, this reviewer does have some major concerns that need to be addressed to strengthen the findings of the manuscript and support the stated conclusions.

Major points.

1. Statistical methods and details are missing throughout much of the manuscript. It is unclear how many cohorts of animals are used for Figure 1-2. This limits the ability of the reader to properly assess the authors' conclusions.
2. Insufficient details are provided for strains of mice used in these studies. The authors state that the AgRP^{Cre}-Ai14 mouse is recently developed but do not provide a citation detailing the origin of these mice. Control mice in most experiments are marked as wildtype (WT) in figures, but it seems more accurate to represent control mice as the CRE parent line.
3. The breeding strategy for generating miR-33^{AgRP}-iKO mice and controls is unclear. Are control mice used in these studies derived only from the parent Cre line without genetic contribution from the miR-33^{loxP/loxP} strain? Although small effect sizes are often expected in microRNA research, small changes are also prevalent between strains and even sub-strains of C57BL/6 mice. If control mice are sourced only from the parent Cre strain it is advisable to also evaluate data for matched miR-33^{loxP/loxP} (negative Cre) mice as a control.
4. The authors claim that miR-33^{AgRP}iKO mice have metabolic dysfunction. However, this appears to be primarily based on impaired glucose clearance following a GTT that has limitations (discussed below). Is there additional data to support metabolic dysfunction? The graphic of the final figure indicates data on dyslipidemia, hepatic steatosis and hyperglycemia, but this data is not presented.
5. Does aging reveal an obesity phenotype in mice fed the chow diet? Being that miR-33 knockout mice in Price et al. 2018 (Cell Reports) developed obesity and metabolic dysfunction within 20 weeks on a chow diet this seems critical for stating that '...specific loss of miR-33 in AgRP neurons is sufficient to mimic the effects of global miR-33 deficiency'.
6. Lean mass is responsible for most glucose clearance in a glucose tolerance test and lean mass is often only minimally increased in diet-induced obesity models (McGuinness et al. 2009; PMID: 19638507). The GTT performed in Fig. 1c/f were normalized to total body weight according to methods. If lean mass is equal between groups, but the miR-33^{AgRP}iKO mice received a larger bolus of glucose (based on increased total mass) it is not surprising that these mice have delayed glucose

clearance in GTT. Moreover, the data would not support metabolic dysfunction in miR-33^ΔAgRPiKO mice. Does lean mass differ between groups? The authors should consider also reporting data as % fat (fat mass/total mass) and % lean (lean mass/total mass). Reporting the data of total mass gained at the end of the study from baseline should also be considered.

7. An impaired glucose tolerance test in diet-induced obesity mouse models is usually attributable to insulin resistance. Are any data on insulin resistance available to support the GTT results of Fig. 1 c,f?

8. Figure 2 is poorly defined and described within the text. It is unclear whether these are male or female mice. It is unclear the age of these mice. It is unclear whether these mice have been fed a chow diet or HFD, and for how long. There is no statistical analysis provided to support that the knockout mice are indeed heavier (per text). Moreover, the data seem to be bimodal for mass in the knockout group. Although intriguing, the ambiguity behind this dataset leads to many questions in context to the larger study.

9. With regard to Figure 2d-f, variability in body composition complicates calculations of energy expenditure (Kaiyala et al. 2010; PMID: 20413511). Does accounting for variability in fat mass and lean mass in Fig. 2d-f maintain the conclusion that miR-33^ΔAgRPiKO mice are below expectations for energy expenditure?

10. The single-cell sequencing dataset is very fascinating and supportive, but only briefly described. For the sake of transparency, could the authors please provide the number of cells each genotype contributed to the neuron cluster (i.e., is the 10,000 cell sampling strategy providing roughly equal numbers of neurons). Additionally, please reference how many neurons of the total neuron pool were identified as AgRP+. It appears that a differential expression analysis was performed to compare control and miR-33KO cells identified as AgRP+. What statistical tests/software were used to support changes in expression between genotypes? This differential expression dataset should be disclosed as supplemental material with normalized expression level, fold-change and statistic.

Minor comments:

1. There are typos throughout the manuscript. Proofread carefully.

2. Fig. 1–a,d. Unclear what is being compared here statistically – final time point? Slope? It appears that both male and female miR-33^ΔAgRPiKO mice are larger than control mice at time 0. Plotting the change in mass from start to end may be more useful.

3. Figure-1i,j. It is unclear whether these mice are fed a chow diet or HFD before and after fasting.

4. Figure-1i,j. The figure legend indicates 24h fasting, but text indicates overnight fasting. This is an important distinction.

5. Supplemental Fig 2f-i. While the experiments in S2f-I have merit, is it appropriate to draw conclusions with n=3 for the control group? At the minimum, please address this limitation in the text

6. Figure 3a- There is no scale bar.

7. Figure 3b-e – Area is represented as ml/hr. Y-axis is missing units for others.

8. Figure 3g – Ampk is listed as a transcript. Please correct to Prkaa1.

9. Figure 4b-c. Could the authors please state the % mitochondrial read cutoffs used for filtering data in the methods.

10. Figure 4c-f. – The figure legend indicates that a differential expression analysis was performed

between AgRP+ cells of each genotype. Please include this data along with methods for this statistical analysis. Does the differential expression analysis support that AgRP expression is increased in miR-33^{-/-}AgRPiKO as referenced? What about genes of e, f? If depth limits this analysis, please clarify.

11. Figure 4c. Please label axes as UMAP1/2 with scale.

12. Figure 4d. There is no x-axis scale, label, or statistic.

13. Figure 4g – cFos is labeled 'Fos'. Also missing scale bar.

14. Figure 4h. The bottom of the figure indicates that the miR-33^{-/-}AgRPiKO has increased food intake leading to obesity with increased adiposity, fatty liver, dyslipidemia and hyperglycemia. This manuscript supports a modest increase in adiposity, but no data is shown to support increased fatty liver, dyslipidemia or hyperglycemia in the miR-33^{-/-}AgRPiKO. Please revise or provide data to support these metabolic dysfunctions.

15. The methods mention 'WD feeding'. Please clarify if a diet other than HFD / chow was used.

Reviewer #2 (Remarks to the Author):

In the present study, the authors found that miR-33 plays an important role in the regulation of hypothalamic AgRP neuron activity. Specific removal of miR-33 from these cells significantly increased several miR-33 target genes, including those involved in mitochondrial function and long-chain fatty acid metabolism, and increased AgRP cell activity. These results suggest that miR-33 may play an important organizational role in suppressing the activity of AgRP neurons and thereby suppressing hunger and the development of obesity and metabolic dysfunction through the combined regulation of these related factors. While these are interesting results, the following points need to be clarified.

Major

1. In the specific loss of miR-33 in AgRP neurons, there is a change in food intake and body weight during a high-fat diet load, but no difference in food intake on a normal diet. This is also the case in previous papers. On the other hand, in previous papers, body weight differences were also observed in the normal diet. This suggests that the effect of miR-33 on AgRP neurons may be limited, if any.

2. As the authors point out, it has been reported in the past that the activation of the sympathetic nervous system is reduced when miR-33 is deficient in catecholamine-producing cells. This may be the reason for the difference in body weight despite the same amount of food consumed on a normal diet.

3. miR-33-deficient AgRP neurons have been shown to have elevated miR-33 target genes previously reported, but no novel target genes have been demonstrated. Concurrently, mitochondrial density has been shown to be elevated, but no causal relationship has been demonstrated as to whether changes in gene expression are directly related to mitochondrial changes or to feeding behavior.

Reviewer #3 (Remarks to the Author):

In this study, Price and collaborators investigate the role of miR-33 in brain cell types implicated in appetite and metabolic control. In particular, the authors generated 3 mouse models with conditional deletion of miR-33 in astrocytes, POMC and AgRP neurons. Only mice lacking miR-33 in AgRP neurons exhibited a metabolic phenotype, characterized by hyperphagia, increased body weight and altered glucose metabolism under obesogenic conditions. Using bulk AgRP neuron-specific sequencing (using

the Ribotag model) and single-cell sequencing, the authors found that this phenotype was associated with alterations in the expression of miR-33 target genes involved in mitochondrial biogenesis and fatty acid metabolism. Together, the authors suggest that these changes would eventually cause a permanent activation of AgRP neurons and an increase in the expression of orexigenic neuropeptides thus leading to overweight.

The authors used a combination of mouse genetics, physiological studies, electron microscopy and RNA sequencing to draw their conclusions. Generally speaking, the study is interesting and well performed. However, there are some questions that should be addressed:

1. There is no Statistical analysis section. This is an important omission that impedes to properly review the data of the manuscript.
2. The genetic ablation of miR-33 in the different cell types is not demonstrated. While these Cre-lines tend to be very effective in recombining the floxed gene, it is a good scientific practice to show the extent of deletion of the target gene.
3. The authors only show the phenotype of miR33-AgRPKO mice under HFD conditions. I assume that this is because under standard diet there is no phenotypical alterations. If so, this should be stated in the text and shown basic parameters (i.e., body weight, glucose metabolism) as a supplementary information.
4. Fig 1g-h. The authors mentioned that they did continuous monitoring of food intake but only showed total food consumption. Patterns of food intake should be shown as they can be very informative.
5. Fig 1k indicates that KO mice are more sensitive to the orexigenic effects of ghrelin. Are basal circulating levels of ghrelin increased? Or AgRP neurons have increased expression of ghrelin receptors? Is this a cell autonomous effect?
6. In Fig 4c, the authors state that AgRP expression is increased. The authors should provide statistics in this graph.
7. Fig 4f and e seems to lack statistics as well.
8. Overall, the data presented suggest that AgRP neurons lacking miR-33 have increased fatty acid oxidation, increased activity and expression of AgRP/NPY neuropeptides. Is this elevated activity state of AgRP neurons permanent? Or is more obvious during light phase (associated with fasting) when AgRP neurons are usually more active? Is food intake increased during the light phase (see question #3)?
9. I assume that Fig 4g is analysed under basal (fed) conditions in view of the low number of active AgRP neurons. This should be stated. Since the authors explain the whole phenotype of the mutant mice by an enhanced activity of AgRP neurons, perhaps it would be good to use a second way to confirm that AgRP neurons are activated (i.e., fibre photometry).

Other questions:

1. The introduction should be framed better in relation to the central/hypothalamic control of appetite and energy balance. For example, the authors should introduce the cell types investigated (astrocytes, POMC and AgRP neurons), explain their role in energy balance control and justify why target them.
2. Body weight graphs in Fig 1a and d should be properly discontinued between days 28-42 and 77-98.
3. In general, the text does not provide too much information about the conditions in which the experiments have been done. When relevant, it should be stated if the studies are under chow/HFD, fed/fasted, light/dark phase, etc.
4. Line 123: wrong title.
5. Fig 1a and b. The breeding strategy should be indicated by "x" rather than "-" (i.e. miR-33lox/lox x AgRPCre-ERT2).
6. Fig 1g and h. Why data on chow diet is shown if only HFD is included in the manuscript? Basic phenotypical data on chow diet should be provided (see point 2).
7. Fig 2 d-f: is this data analysed by ANCOVA?
8. Fig 4a: the figure is too small and barely readable.

9. Fig 4g. Colocalization should be indicated by arrows. The FOS staining is difficult to appreciate.
10. Fig 4h. In the lower panel, the authors state some alterations that have not been demonstrated in this study (i.e., fatty liver, dyslipidaemia). Hyperglycaemia is not shown in Fig 1, although these mice have alterations in glucose metabolism. This panel should be either removed or listed with the metabolic alterations for which the authors provide evidence.
11. Lines 120-121. The authors suggest that the specific loss of miR33 in AgRP neurons recapitulates the global deficiency of miR33. Nevertheless, it seems that the body weight difference in the global KO (refs 11, 12) is much larger than the one reported in the present study. In ref 11 the authors even show a notable difference in body weight under chow diet conditions. While these may be due to differences in genetic strain, animal house conditions (i.e. gut microbiota), etc. this statement should be toned-down and further discussed.

RESPONSE TO The REVIEWERS

Reviewer #1 (Remarks to the Author):

The scope of the manuscript by Price et al. is likely appropriate for Nature Communications. It is a brief, but well-written, report on the role of miR-33 as a regulator of food-seeking behavior. Over-nutrition is a key component of the obesity epidemic that puts increasing strain on public health. Reversing obesity requires lifestyle interventions to include exercise and caloric restriction. However, the molecular mechanisms that regulate hunger are not fully defined. The present study adds detail to this knowledge gap by reporting that mice lacking miR-33 expression specifically in AgRP⁺ neurons have increased food consumption and obesity that is perhaps linked to enhanced mitochondrial biogenesis and function in AgRP⁺ neurons. The latter is supported by innovative and independent models of transcriptional profiling of AgRP⁺ neurons.

Conceptually, the story is interesting and stimulates new questions on the molecular mechanisms that regulate hunger and over-nutrition. The introduction is succinct and mostly straight-forward, but an improved transition between the last sentence to the results may help direct the reader. The discussion is also concise and direct. Although the results look solid, this reviewer does have some major concerns that need to be addressed to strengthen the findings of the manuscript and support the stated conclusions.

Major points.

1. “Statistical methods and details are missing throughout much of the manuscript. It is unclear how many cohorts of animals are used for Figure 1-2. This limits the ability of the reader to properly assess the authors’ conclusions”.

We apologize for the limited details on statistical analyses and have provided additional details throughout the manuscript. For Figure 1 body weight and GTTs shown in panels B-F (new version Figure 1 a-j) as well as for new data included in new Figure 2 were based on combined data from three different cohorts that all showed similar differences between control and miR-33^{AgRPiKO} mice. Panels g-k (new version Figure 3) as well as Figure 2 (new version Figure 4) were derived from separate individual cohorts of animals that were single housed to perform assessments of food consumption and respiratory parameters in individual animals.

2. “Insufficient details are provided for strains of mice used in these studies. The authors state that the AgRP^{Cre}-Ai14 mouse is recently developed but do not provide a citation detailing the origin of these mice. Control mice in most experiments are marked as wildtype (WT) in figures, but it seems more accurate to represent control mice as the CRE parent line”.

We regret not having not providing further details on the mouse strains utilized in this work. While the adult inducible AgRP conditional line used in this work was newly developed at the time this study was initially performed, it have since been used in other work and additional details as well as a citation of this earlier work have been included. Briefly, mice expressing a tamoxifen-inducible Cre recombinase (CreERT2) in cells expressing AgRP (Agrp^{Cre:ERT2}, Wang et al. 2014; PMID: PMC3929914) were crossed with Rosa26-lox-stop-lox-tdTomato (Ai14; cre-recombinase-dependent expression) mice (Ai14 reporter mice; stock #007914; The Jackson Laboratory, Bar Harbor, ME, USA) to label AgRP-expressing cells. AgRP-ERT2^{Cre-Ai14} mice have

AgRP-positive cells that express tdTomato upon tamoxifen administration. No observation of AgRP-tdTomato expression was found in the absence of tamoxifen administration, indicating that recombination was strictly dependent upon tamoxifen-induced Cre recombinase activation (Jin S. et al. 2021; PMID: PMC7946429). miR-33^{loxP/loxP} mice were crossed with AgRP-ERT2^{Cre-Ai14} mice to generate mice that were heterozygous for the miR-33 inducible knockout allele, with and without AgRP-ERT2^{Cre-Ai14}. These mice were then intercrossed to generate mice homozygous for the miR-33^{loxP/loxP} or WT allele with AgRP-ERT2^{Cre-Ai14}.

3. “The breeding strategy for generating miR-33^{AgRP-iKO} mice and controls is unclear. Are control mice used in these studies derived only from the parent Cre line without genetic contribution from the miR-33^{loxP/loxP} strain? Although small effect sizes are often expected in microRNA research, small changes are also prevalent between strains and even sub-strains of C57BL/6 mice. If control mice are sourced only from the parent Cre strain it is advisable to also evaluate data for matched miR-33^{loxP/loxP} (negative Cre) mice as a control”.

We apologize for not fully and explicitly describing the breeding strategy used for generating miR-33^{AgRPiKO} mice and other conditional knockout lines used in this study. In the revised version of the manuscript a detailed description of the breeding strategies used to generate miR-33^{AgRPiKO} mice as well as the other strains described in this manuscript has been provided in the methods section. As all mice utilized are from the C57BL/6J strain and both the Cre strain and miR-33^{loxP/loxP} strain contribute equally to both control and miR-33^{AgRPiKO} mice, any possible differences in sub-strains should not contribute to the observed differences.

4. “The authors claim that miR-33^{AgRPiKO} mice have metabolic dysfunction. However, this appears to be primarily based on impaired glucose clearance following a GTT that has limitations (discussed below). Is there additional data to support metabolic dysfunction? The graphic of the final figure indicates data on dyslipidemia, hepatic steatosis and hyperglycemia, but this data is not presented”.

The reviewer rightly points out that the graphical abstract indicates changes that were not demonstrated in the data provided. In the revised version we have provided additional data to support the information shown in the graphical abstract. In order to more fully determine which of the metabolic deficiencies observed global miR-33 KO mice were observed in our miR-33^{AgRPiKO} mice, we have performed measurements of triglycerides in the liver as well as triglycerides and cholesterol in plasma collected from these animals. These measurements demonstrate that miR-33^{AgRPiKO} mice have increased triglyceridemia as shown by the higher levels of triglycerides in serum of both males and females after high fat diet. On the other hand, serum cholesterol and HDL-cholesterol levels were moderately reduced in males, while an increase in total cholesterol was observed in female mice. Serum levels related to triglyceride and cholesterol are included in new Figure 2 a-f. Additionally, hepatic steatosis evaluation demonstrates that miR-33^{AgRPiKO} mice accumulate greater amounts of triglycerides and cholesterol in the liver (including total cholesterol and free cholesterol) than WT mice upon high fat diet feeding. This was observed in both males and females. These new data reflecting increased lipid accumulation in the liver are shown in Figure 2 g-l. This new data is similar to the metabolic alterations observed in the global miR-33 KO mice, including hypertriglyceridemia and hepatic steatosis.

5. “Does aging reveal an obesity phenotype in mice fed the chow diet? Being that miR-33 knockout mice in Price et al. 2018 (Cell Reports) developed obesity and metabolic dysfunction within 20 weeks on a chow diet this seems critical for stating that ‘...specific loss of miR-33 in AgRP neurons is sufficient to mimic the effects of global miR-33 deficiency”.

Unfortunately, we have not evaluated whether miR-33^{AgRPiKO} develop obesity and metabolic dysfunction when fed a chow diet for a sustained period of time. As we were evaluating the impact of selective loss of miR-33 in a number of different tissues and cell types we focused upon the phenotypes that were observed in response to HFD feeding, which were more robust and manifested more rapidly. We have modified the revised version of the manuscript to more clearly state that loss of miR-33 in AgRP neurons was sufficient to largely mimic the effects of global miR-33 deficiency in response to HFD feeding. Additionally, we have included in the revised version of the manuscript a section on “Study limitations” wherein we specify that the long-term effects of specific loss of miR-33 in AgRP neurons was not evaluated in chow fed animals (lines 327-347).

6. “Lean mass is responsible for most glucose clearance in a glucose tolerance test and lean mass is often only minimally increased in diet-induced obesity models (McGuinness et al. 2009; PMID: 19638507). The GTT performed in Fig. 1c/f were normalized to total body weight according to methods. If lean mass is equal between groups, but the miR-33^{AgRPiKO} mice received a larger bolus of glucose (based on increased total mass) it is not surprising that these mice have delayed glucose clearance in GTT. Moreover, the data would not support metabolic dysfunction in miR-33^{AgRPiKO} mice. Does lean mass differ between groups? The authors should consider also reporting data as % fat (fat mass/total mass) and % lean (lean mass/total mass). Reporting the data of total mass gained at the end of the study from baseline should also be considered”.

We thank the reviewer for these excellent suggestions. In both our prior work on global miR-33 deficiency as well as this work, the amount of glucose administered during the GTT was adjusted according to the body weight of these animals. As such, the impaired glucose clearance during GTT's in both the whole body and AgRP specific miR-33 knockout mice was likely a result of the increased body weight of these animals. While this approach does have limitations, it is still the recommended protocol for performing GTTs. As suggested, we have included data on lean mass in the revised version of the manuscript (Figure 1b right panel and Figure 1d right panel) This data shows that male miR-33^{AgRPiKO} mice had a significant increase in lean mass as well as fat mass, while in females a similar trend was observed, but this was not found to be significant. Consistent with this, the percent lean mass and percent fat mass were not significantly different between groups in male mice, but a significant increase in relative fat mass and a trend to a reduction in relative lean mass were observed in females. As the percent lean mass does differ between groups in females, we have added a discussion about how this change in body composition could contribute to the effects observed in GTTs (lines 288-292), although this is unlikely to be the case in males where no substantial differences in % body composition were observed (Supplemental Figure 1c-d). In the revised manuscript we have also reported data for change in BW from initiation of HFD (Supplemental Figure 1a-b). Similar to what we observed for total body mass, the increase in body weight following HFD feeding was also significantly higher for both male and female miR-33^{AgRPiKO} mice.

7. “An impaired glucose tolerance test in diet-induced obesity mouse models is usually attributable to insulin resistance. Are any data on insulin resistance available to support the GTT results of Fig. 1 c,f?”

In addition to GTT's, we also performed ITT's in some of the cohorts used in this study. Female miR-33^{AgRPiKO} mice showed a diminished response to insulin. For males, ITT data was only available for 4 animals per group. The response to insulin stimulation was less pronounced than in females and did not show any clear differences. This data has been included in Figure 1 g and i. Additionally, we measured fasting insulin levels in the different cohorts used in the study

after HFD. Similar to ITT, fasted insulin was higher in female miR-33^{AgRPiKO} mice, suggesting increased insulin resistance, while insulin data for males did not show clear differences. This data has been included in Figure 1 h and j.

8. “Figure 2 is poorly defined and described within the text. It is unclear whether these are male or female mice. It is unclear the age of these mice. It is unclear whether these mice have been fed a chow diet or HFD, and for how long. There is no statistical analysis provided to support that the knockout mice are indeed heavier (per text). Moreover, the data seem to be bimodal for mass in the knockout group. Although intriguing, the ambiguity behind this dataset leads to many questions in context to the larger study”.

Figure 2 (new Figure 4) is based on data from male mice after one month on HFD. In this subcohort of animals the difference in body weight was not significant at the time measurements were taken. The text has been modified to state that “Body mass has a major impact on respiratory parameters, and although not significantly different in the subset of animals used in this analysis, miR-33^{AgRPiKO} mice had a trend toward higher average body weight.” (lines 160-163). In this study, as well as the work done with global miR-33 deletion, we observed a high degree of variability in body weight as well as other parameters in both the control and miR-33 deficient animals. This is not unique to this study, but is a common feature of the mouse models used, especially in studies of diet induced obesity. For this reason, these experiments were repeated in three different cohorts of animals to ensure the reproducibility of this phenotype and obtain sufficient numbers to demonstrate significant differences despite the inherent phenotypic variability. While we would have liked to have also performed indirect calorimetry in all of the animals included in this study, this equipment is not owned by our lab and so these measurements were only performed after we had established which of the models we tested was exhibiting a phenotype.

9. “With regard to Figure 2d-f, variability in body composition complicates calculations of energy expenditure (Kaiyala et al. 2010; PMID: 20413511). Does accounting for variability in fat mass and lean mass in Fig. 2d-f maintain the conclusion that miR-33^{AgRPiKO} mice are below expectations for energy expenditure?”

We agree with the reviewer that differences in body composition can impact respiratory parameters. Unfortunately, body composition was not measured in the cohort of animals used for metabolic cage experiments, so it is not possible to assess this directly in these animals. A discussion of how alterations in body composition could contribute to some of these differences has been included in the revised version of the manuscript (line 327-336).

10. “The single-cell sequencing dataset is very fascinating and supportive, but only briefly described. For the sake of transparency, could the authors please provide the number of cells each genotype contributed to the neuron cluster (i.e., is the 10,000 cell sampling strategy providing roughly equal numbers of neurons). Additionally, please reference how many neurons of the total neuron pool were identified as AgRP+. It appears that a differential expression analysis was performed to compare control and miR-33KO cells identified as AgRP+. What statistical tests/software were used to support changes in expression between genotypes? This differential expression dataset should be disclosed as supplemental material with normalized expression level, fold-change and statistic”.

We thank the reviewer for the positive comments on the scRNA-seq and the suggestion to clarify the methodology related to this analysis. The methods section has been modified to provide more information on the workflow process. Briefly, from the 10,000 cells derived from

each genotype, 7891 WT and 8160 miR-33^{AgRPiKO} were neurons. Then, 73 WT and 116 miR-33^{AgRPiKO} AgRP positive neurons were selected for further analysis. To identify differentially expressed genes in AgRP neurons, Seurat FindAllMarkers function with Poisson test for UMI-based dataset was used, a $P < 0.05$ was considered significant. As suggested, we have included the complete list of differentially expressed genes in Supplementary Data 2.

Minor comments:

1. “There are typos throughout the manuscript. Proofread carefully”.

The revised version of the manuscript has been carefully proofread by multiple authors to identify and correct typos and other grammatical issues.

2. “Fig. 1–a,d. Unclear what is being compared here statistically – final time point? Slope? It appears that both male and female miR-33^{AgRPiKO} mice are larger than control mice at time 0. Plotting the change in mass from start to end may be more useful”.

2-way Anova was performed to determine whether there was a significant difference in the body weight across time between control and miR-33^{AgRPiKO} mice following HFD feeding. While there was not a significant difference in body weight between groups at the onset of HFD feeding, miR-33^{AgRPiKO} mice did tend to have slightly higher body weight. When data is plotted as change in mass, we observed that miR-33^{AgRPiKO} gain significantly more weight than control animals (Supplemental Figure 1a-b).

3. “Figure-1i,j. It is unclear whether these mice are fed a chow diet or HFD before and after fasting”.

We apologize for not stating this more clearly in the text and figure legend. The animals were fed a chow diet. As chow diet fed animals did not show differences in daily food consumption under normal conditions, we tested whether challenging these animals with fasting or administration of ghrelin would allow us to observe differences in feeding behavior that were otherwise not apparent.

4. “Figure-1i,j. The figure legend indicates 24h fasting, but text indicates overnight fasting. This is an important distinction”.

We are sorry for this oversight. As stated in the figure legend, mice were fasted for 24 hours prior to assessment of food consumption.

5. “Supplemental Fig 2f-i. While the experiments in S2f-I have merit, is it appropriate to draw conclusions with n=3 for the control group? At the minimum, please address this limitation in the text”

We agree that the number of animals used to assess the phenotype of miR-33^{GFAPIKO} mice was not sufficient to draw any firm conclusions. We have reworded the text to state that initial characterization of a limited number of miR-33^{GFAPIKO} mice did not reveal any apparent differences in body weight compared to control animals, indicating that the low number of animals used in these measurements limits the ability to draw firm conclusions (lines 182-185).

6. “Figure 3a- There is no scale bar”.

Scale bar has been added.

7. “Figure 3b-e – Area is represented as ml/hr. Y-axis is missing units for others”.

Y axis in Figure 3b (new Figure 5b) has been corrected and units for Figures 3c-e (new Figure 5c-e) have been added to axes.

8. “Figure 3g – Ampk is listed as a transcript. Please correct to Prkaa1”.

This has been corrected. Additionally, Pgc1a has been corrected to Ppargc1a.

9. “Figure 4b-c. Could the authors please state the % mitochondrial read cutoffs used for filtering data in the methods”.

We have included the information for the % mitochondrial threshold in the methods section. Mitochondrial cutoffs was set at 40%.

10.” Figure 4c-f. – The figure legend indicates that a differential expression analysis was performed between AgRP+ cells of each genotype. Please include this data along with methods for this statistical analysis. Does the differential expression analysis support that AgRP expression is increased in miR-33^{AgRPiKO} as referenced? What about genes of e, f? If depth limits this analysis, please clarify”.

We have included a more extended description of the differential expression analysis in the methods section. The global effect of miR-33 deficiency in these neurons is represented in the panel referring to the IPA canonical pathways analysis, in which Z-score is represented for pathways with a -Log p>1.3. As described in the methods section, differential gene expression analysis Seurat FindAllMarkers function with Poisson test for UMI-based dataset was used, a P < 0.05 was considered significant. However, as pointed out by the reviewer, the depth of analysis can impair this analysis regarding the expression of particular genes, due the limited number of AgRP positive cells in the analysis and low average gene expression within these cells. We have included the statistical analysis for differential expression between genotypes in the respective figure panel, including AgRP, Npy, and Ppargc1a. However, no statistical significance was found for the expression levels of other genes in E or F based on the scRNAseq analysis. In these two panels, we focus our attention on the higher relative number of cells expressing the indicated genes, which strengthen the phenotype of an increased activation of AgRP neurons in the miR-33^{AgRPiKO}. We have modified the description of this data in the results to more clearly indicate what significant differences were observed. Together we believe that the gene expression pattern derived from this analysis correlates with and supports the gene expression analysis we have performed by standard rt-qPCR in AgRP neurons purified according to the Ribotag pull-down strategy.

11. “Figure 4c. Please label axes as UMAP1/2 with scale”.

We apologize for missing axis labeling. UMAP1/2 axes with appropriate scale have been added.

12. “Figure 4d. There is no x-axis scale, label, or statistic”.

X-axis and scale have been included to reflect Z-score, which was used for the Ingenuity Pathway Analysis. All pathways included were above the threshold of a -Log p>1.3.

13. “Figure 4g – cFos is labeled ‘Fos’. Also missing scale bar”.

We thank the reviewer for pointing this out. Both have been corrected accordingly.

14. “Figure 4h. The bottom of the figure indicates that the miR-33^{AgRPiKO} has increased food intake leading to obesity with increased adiposity, fatty liver, dyslipidemia and hyperglycemia. This manuscript supports a modest increase in adiposity, but no data is shown to support increased fatty liver, dyslipidemia or hyperglycemia in the miR-33^{AgRPiKO}. Please revise or provide data to support these metabolic dysfunctions”

We apologize again for this summary figure. Additional data supporting dyslipidemia and fatty liver in the miR-33^{AgRPiKO} mice has been added to the manuscript (Figure 2), demonstrating that these phenotypes are observed in both global and AgRP neuron specific miR-33 knockout mice.

15. The methods mention ‘WD feeding’. Please clarify if a diet other than HFD / chow was used.

Only the HFD and chow diets described in the methods have been used in this study. The reference to WD feeding has been corrected.

Reviewer #2 (Remarks to the Author):

In the present study, the authors found that miR-33 plays an important role in the regulation of hypothalamic AgRP neuron activity. Specific removal of miR-33 from these cells significantly increased several miR-33 target genes, including those involved in mitochondrial function and long-chain fatty acid metabolism, and increased AgRP cell activity. These results suggest that miR-33 may play an important organizational role in suppressing the activity of AgRP neurons and thereby suppressing hunger and the development of obesity and metabolic dysfunction through the combined regulation of these related factors. While these are interesting results, the following points need to be clarified.

Major

1. “In the specific loss of miR-33 in AgRP neurons, there is a change in food intake and body weight during a high-fat diet load, but no difference in food intake on a normal diet. This is also the case in previous papers. On the other hand, in previous papers, body weight differences were also observed in the normal diet. This suggests that the effect of miR-33 on AgRP neurons may be limited, if any”.

While loss of miR-33 either in the whole body or AgRP neurons did not result in increased feeding in chow diet fed mice under normal conditions, we do see increased feeding in HFD fed mice, as well as chow diet fed mice that are fasted or treated with ghrelin. These findings demonstrate that the ability of the AgRP neurons to respond to multiple different energetic challenges is altered indicating that miR-33 plays an important role in regulating the function of this neuronal sub-type. While this work does not determine whether prolonged feeding with chow diet would result in alterations in body weight and metabolic function, it is clear that in the context of high fat diet feeding the effect of miR-33 on AgRP neurons is sufficient to significantly alter the metabolic health of these animals.

2. “As the authors point out, it has been reported in the past that the activation of the sympathetic nervous system is reduced when miR-33 is deficient in catecholamine-producing

cells. This may be the reason for the difference in body weight despite the same amount of food consumed on a normal diet”.

As the reviewer suggests, the altered adaptive thermogenesis and reduced energy expenditure observed in mice lacking miR-33 in Dbh-positive cells could contribute to the increased body weight and impaired metabolic function of miR-33 deficient animals. However, no body weight phenotype was reported in these mice, suggesting that the effect of miR-33 in Dbh-positive cells was not sufficient to promote the obesity phenotype observed in global miR-33 knockout mice. While AgRP neuron specific loss of miR-33 does result in a similar phenotype to that observed in global miR-33 knockout mice in response to HFD, the effects are in some ways less pronounced, suggesting that additional roles of miR-33 in Dbh-positive neurons or other cell types may also contribute to this phenotype. In the revised version of the manuscript further discussion of this is provided (lines 289-297).

3. “miR-33-deficient AgRP neurons have been shown to have elevated miR-33 target genes previously reported, but no novel target genes have been demonstrated. Concurrently, mitochondrial density has been shown to be elevated, but no causal relationship has been demonstrated as to whether changes in gene expression are directly related to mitochondrial changes or to feeding behavior”.

As AgRP neurons make up only a small proportion of the cells within the arcuate nucleus, we have applied a combination of single cell sequencing and ribosomal pulldown of Cre-positive cells to allow us to assess the impact of miR-33 deficiency in AgRP neurons. As the depth of coverage in single cell sequencing from rare cellular subtypes, as well as the amount of material obtained from our RiboTag mouse model was limited, our initial focus was upon established miR-33 target genes that have previously been shown to regulate the activity of AgRP neurons. We were able to demonstrate that the number of AgRP neurons expressing these miR-33 target genes was increased in cells lacking miR-33 as was mitochondrial density, which is regulated by some of these miR-33 targets and has also been linked to the function of these neurons. As miR-33, like other microRNAs, has the ability to target many different mRNAs, other factors are likely also be involved in the ability of miR-33 to mediate these effects. In the revised version of the manuscript, we have performed a more extensive analysis of our single cell data and identified additional predicted miR-33 target genes that are dysregulated in AgRP neurons lacking miR-33. The analysis was performed based on overlapping genes predicted to be miR-33 targets by TargetScan7.2 and genes significantly ($p < 0.05$) upregulated in AgRP neurons as determined from our scRNA-seq data using the Seurat FindAllMarkers function with Poisson test for UMI-based dataset. Interestingly, novel miR-33 target genes were found to be upregulated in the AgRP neurons lacking miR-33, including genes related to lipid metabolism, GABAergic function and calcium signaling, functions that have been previously linked to AgRP activation. We have included these results in Figure 6h and Supplementary Data 3. In the future it will be important to assess whether these genes are directly regulated by miR-33 and if their regulation by miR-33 plays a direct role in AgRP neuronal activity.

Reviewer #3 (Remarks to the Author):

In this study, Price and collaborators investigate the role of miR-33 in brain cell types implicated in appetite and metabolic control. In particular, the authors generated 3 mouse models with conditional deletion of miR-33 in astrocytes, POMC and AgRP neurons. Only mice lacking miR-33 in AgRP neurons exhibited a metabolic phenotype, characterized by hyperphagia, increased

body weight and altered glucose metabolism under obesogenic conditions. Using bulk AgRP neuron-specific sequencing (using the Ribotag model) and single-cell sequencing, the authors found that this phenotype was associated with alterations in the expression of miR-33 target genes involved in mitochondrial biogenesis and fatty acid metabolism. Together, the authors suggest that these changes would eventually cause a permanent activation of AgRP neurons and an increase in the expression of orexigenic neuropeptides thus leading to overweight.

The authors used a combination of mouse genetics, physiological studies, electron microscopy and RNA sequencing to draw their conclusions. Generally speaking, **the study is interesting and well performed**. However, there are some questions that should be addressed:

1. “There is no Statistical analysis section. This is an important omission that impedes to properly review the data of the manuscript”.

We apologize for this oversight. In the revised version of the manuscript, we have included a statistical analysis section in the materials and methods and provided further information on the types of statistics used in the figure legends.

2. “The genetic ablation of miR-33 in the different cell types is not demonstrated. While these Cre-lines tend to be very effective in recombining the floxed gene, it is a good scientific practice to show the extent of deletion of the target gene”.

We have previously demonstrated the efficacy and selectivity of our conditional miR-33 knockout allele in other tissues (Price et al. 2021; PMCID: PMC7865172), and prior work has also demonstrated the specificity of the AgRP-cre model used (Jin S. et al. 2021; PMCID: PMC7946429). Unfortunately, neither the RiboTag or single cell sequencing approaches employed in this work allow us to evaluate the expression of miRNAs, so we were not able to directly assess miR-33 expression in these cells although the alterations in the activity and functions of these neurons as well as the alterations in miR-33 target genes do indicate effective removal of miR-33 from these neurons. A discussion of this has been added as a limitation of this work (lines 340-345).

3. “The authors only show the phenotype of miR33-AgRPKO mice under HFD conditions. I assume that this is because under standard diet there is no phenotypical alterations. If so, this should be stated in the text and shown basic parameters (i.e., body weight, glucose metabolism) as a supplementary information”.

Mice globally deficient for miR-33 did not show an overt phenotype at a young age. Similar to this, miR-33^{AgRPiKO} mice were not found to have substantial differences in body weight at a young age. A statement to this effect has been added to the revised manuscript (lines 114-116). As described in response to Point 5 of reviewer one, we did not perform an extensive characterization of miR-33^{AgRPiKO} mice fed a chow diet and have discussed this a limitation of this study in the revised version of the manuscript (lines 328-334).

4. “Fig 1g-h. The authors mentioned that they did continuous monitoring of food intake but only showed total food consumption. Patterns of food intake should be shown as they can be very informative”.

We apologize for this misstatement. The data shown in Fig 1g-h was not based on continuous monitoring of food intake, but from regular measurements of food consumption every 1-3 days after initiation of HFD feeding. Further assessments of feeding behavior over

shorter time periods in animals that were either fasted for 24 hours or treated with ghrelin were also performed, but the type of continuous monitoring of food intake needed to assess circadian differences in feeding patterns were not collected.

5. “Fig 1k indicates that KO mice are more sensitive to the orexigenic effects of ghrelin. Are basal circulating levels of ghrelin increased? Or AgRP neurons have increased expression of ghrelin receptors? Is this a cell autonomous effect?”

We have measured circulating ghrelin in fasted male and female mice after HFD feeding and did not find any significant changes between genotypes. This data has been included in Supplementary Figure 1e). We also attempted to check ghrelin receptor expression in our scRNA-seq data but were not able to detect it at significant levels.

6. “In Fig 4c, the authors state that AgRP expression is increased. The authors should provide statistics in this graph”.

AgRP expression statistic has been added to the graph (new Figure 6c). A more detailed description of the methods used to analyze differential gene expression and statistical analysis in the single cell data has been included in the methods section.

7. “Fig 4f and e seems to lack statistics as well”.

Due to the small proportion of AgRP cells in the arcuate nucleus and the limited depth of coverage of scRNA-seq, statistical analysis of gene expression levels was only significant for Npy and Ppargc1a. However, the number of AgRP neurons expressing these genes was found to be higher in the absence of miR-33. As explained in detail in response to Reviewer 1, minor point 10, we modified our description of these results and clarified how the depth of the analysis and small cell number limits the statistical analysis of this section.

8. “Overall, the data presented suggest that AgRP neurons lacking miR-33 have increased fatty acid oxidation, increased activity and expression of AgRP/NPY neuropeptides. Is this elevated activity state of AgRP neurons permanent? Or is more obvious during light phase (associated with fasting) when AgRP neurons are usually more active? Is food intake increased during the light phase (see question #3)?”

While feeding data was not available for mice run in metabolic cages and our measurements of daily feeding rates does not allow us to properly assess circadian rhythms of feeding behavior, the reduced RER observed in miR-33^{AgRPiKO} mice is consistent with a higher relative utilization of fatty acids for energy production. These differences were only observed during the light cycle, suggesting that the impact of miR-33 deficiency is potentially more pronounced during the light phase when AgRP neurons would be expected to be more active.

9. “I assume that Fig 4g is analysed under basal (fed) conditions in view of the low number of active AgRP neurons. This should be stated. Since the authors explain the whole phenotype of the mutant mice by an enhanced activity of AgRP neurons, perhaps it would be good to use a second way to confirm that AgRP neurons are activated (i.e., fibre photometry)”.

The reviewer is correct that this analysis was performed under fed conditions. While we agree that additional direct measurements of AgRP neuronal activity would be ideal, the disruption caused by the Covid19 pandemic derailed our plans to perform these experiments. However, as the knockout of miR-33 in this model is specific to AgRP neurons, we feel that the

changes in feeding behavior, response to HFD, and expression of AgRP and NPY all serve to support the data we provide demonstrating that activity of these neurons is increased.

Other questions:

1. “The introduction should be framed better in relation to the central/hypothalamic control of appetite and energy balance. For example, the authors should introduce the cell types investigated (astrocytes, POMC and AgRP neurons), explain their role in energy balance control and justify why target them”.

A more extensive introduction including the specific information requested by the reviewer has been included in the revised manuscript (lines 86-98).

2. “Body weight graphs in Fig 1a and d should be properly discontinued between days 28-42 and 77-98”.

Due to extenuating circumstances data from some cohorts were not collected at all timepoints. As such these datapoints were not included in the graphs of body weight trajectories. The spacing between datapoints has been adjusted to properly indicate the amount of time that elapsed between the measurements.

3. “In general, the text does not provide too much information about the conditions in which the experiments have been done. When relevant, it should be stated if the studies are under chow/HFD, fed/fasted, light/dark phase, etc”.

We apologize for not providing sufficient details on the conditions in which experiments were performed and data was collected. The revised version of the manuscript provides more information on experimental conditions.

4. “Line 123: wrong title”.

We thank the reviewer for pointing out this mistake. It has been corrected.

5. “Fig 1a and b. The breeding strategy should be indicated by “x” rather than “-“ (i.e. miR-33lox/lox x AgRPCre-ERT2)”.

This has been corrected as suggested.

6. “Fig 1g and h. Why data on chow diet is shown if only HFD is included in the manuscript? Basic phenotypical data on chow diet should be provided (see point 2)”.

Our goal in this work was to test the basic phenotypic effects of whole-body knockout mice in numerous conditional knockout models, including miR-33^{AgRPiKO} mice to better understand how miR-33 is regulating feeding and the development of diet induced obesity. Similar to the global knockout, no difference in daily feeding under chow diet fed conditions was observed. However, we do demonstrate that when challenged with HFD feeding, fasting, or administration of ghrelin mice lacking miR-33 in AgRP neurons have increased feeding. While a full characterization of the impact of AgRP specific loss of miR-33 on body weight changes with age in the absence of HFD was not performed, we do find that the most robust phenotypes of HFD induced weight gain and metabolic dysfunction observed in global miR-33 knockout mice are also observed in this model.

7. “Fig 2 d-f: is this data analysed by ANCOVA?”

Yes. ANCOVA analysis of respiratory parameters was performed. This has now been explicitly stated in the text, methods, and figure legend. A table of statistics from the metabolic cage data has been included as Supplementary Table 1.

8. "Fig 4a: the figure is too small and barely readable".

This figure has been adjusted to make the text more legible.

9. "Fig 4g. Colocalization should be indicated by arrows. The FOS staining is difficult to appreciate".

As suggested arrows have been added to indicate cells showing colocalization.

10. "Fig 4h. In the lower panel, the authors state some alterations that have not been demonstrated in this study (i.e., fatty liver, dyslipidaemia). Hyperglycaemia is not shown in Fig 1, although these mice have alterations in glucose metabolism. This panel should be either removed or listed with the metabolic alterations for which the authors provide evidence".

We apologize for the overstatements of this figure. We have provided additional data on these phenotypes demonstrating that mice lacking miR-33 in AgRP neurons have higher levels of triglycerides and cholesterol both in the liver and in circulation. This data is shown in the new Figure 2.

11. "Lines 120-121. The authors suggest that the specific loss of miR33 in AgRP neurons recapitulates the global deficiency of miR33. Nevertheless, it seems that the body weight difference in the global KO (refs 11, 12) is much larger than the one reported in the present study. In ref 11 the authors even show a notable difference in body weight under chow diet conditions. While these may be due to differences in genetic strain, animal house conditions (i.e. gut microbiota), etc. this statement should be toned-down and further discussed".

We agree and have modified our statements regarding the extent that AgRP specific loss of miR-33 is able to recapitulate the effects of global deficiency and provided further discussion on this topic (lines 294-302).

REVIEWER COMMENTS

Reviewer #1 (Remarks to the Author):

Authors, please accept my compliments on this well executed revision. All comments were answered to this reviewer's satisfaction. I have the following typographical recommendations:

Abstract:

Line 39 – "... target genes involved *in* mitochondrial biogenesis and fatty acid metabolism."

Introduction:

Line 85 – obesity is misspelled

Results:

Line 115 – "...compared *to* control..."

Line 179 – "also" is misspelled in the non-track changes version.

Line 193 – Should the transcript be identified as "Cpt1a"?

It seems the "Discussion" section should actually start on line 233, which is currently the last paragraph of the "Results" section.

Discussion:

Line 300 – Ppargc1a or PGC1a

Figures:

Fig-1h,1j - there is a space between mu and g.

Fig-2c,2f - should read "HDL-C"

Fig-2a,2b,2d,2e - should read "mg/dL"

Fig-3c - colors in legend do not match data.

Fig-4a,4b,4d,4e - should read mL/hr

Reviewer #2 (Remarks to the Author):

The authors have not answered the reviewer's third question. Specifically, the authors show that suppression of miR-33 in AgRP neurons increases the expression of already known targets such as AMPK, CPT1a, and CROT, but they still cannot show whether these increases are really linked to changes in mitochondrial function or feeding behavior.

They also list candidate novel target genes, but have not been able to show what the significance of these elevations might be.

Reviewer #3 (Remarks to the Author):

My concerns have been adequately addressed by the authors, and I have no additional comments. I appreciate the authors for their efforts in resolving my concerns.

RESPONSE TO THE REVIEWERS

Reviewer #1 (Remarks to the Author):

We would like to thank the reviewers for her/his kind words *“Authors, please accept my compliments on this well executed revision. All comments were answered to this reviewer’s satisfaction”*.

Minor editing points:

1. I have the following typographical recommendations:

Abstract:

Line 39 – “... target genes involved **in** mitochondrial biogenesis and fatty acid metabolism.”

Introduction:

Line 85 – obesity is misspelled

Results:

Line 115 – “...compared **to** control...”

Line 179 – “also” is misspelled in the non-track changes version.

Line 193 – Should the transcript be identified as “Cpt1a”?

It seems the “Discussion” section should actually start on line 233, which is currently the last paragraph of the “Results” section.

Discussion:

Line 300 – Ppargc1a or PGC1a

Figures:

Fig-1h,1j - there is a space between mu and g.

Fig-2c,2f - should read "HDL-C"

Fig-2a,2b,2d,2e - should read "mg/dL"

Answer: All the edits and suggestions have been included in the revised version of the manuscript

Reviewer #2 (Remarks to the Author):

The authors have not answered the reviewer's third question. Specifically, the authors show that suppression of miR-33 in AgRP neurons increases the expression of already known targets such as AMPK, CPT1a, and CROT, but they still cannot show whether these increases are really linked to changes in mitochondrial function or feeding behavior. They also list candidate novel target genes but have not been able to show what the significance of these elevations might be.

Answer: In response to the reviewer’s original comment, we performed a more in-depth analysis of our single cell sequencing data to identify additional predicted miR-33 targets that are upregulated in AgRP neurons lacking miR-33 and state that these and other as yet identified miR-33 targets may also be responsible for mediating some of the observed effects. The reviewer is however correct that we were not able to directly demonstrate which of the upregulated miR-33 targets is primarily responsible for mediating the effects of miR-33 on AgRP neurons. We have now provided further modifications to the manuscript to more explicitly state our inability to perform this assessment and discuss the challenges of demonstrating the effects of individual miRNA targets. Specifically, we discuss how miRNAs often mediate their effects through targeting of multiple mRNAs in the same or related pathways. We further assert that “this type of assessment would require the generation of multiple novel mouse models in which the ability of miR-33 to target individual mRNAs, and likely a combination of different mRNAs, is disrupted. While our lab is one of the only labs to have generated this type of binding site mutant mouse model and the only ones to use this approach to demonstrate that a specific miRNA target is directly responsible for mediating the effect of a miRNA on a complex physiologic phenotype in vivo, it is a challenging undertaking and unlikely to produce the desired impact if multiple different

targets are involved in mediating the effects. Furthermore, to the best of our knowledge this has never been performed in a cell type specific manner as would be required to assess the specific effects in AgRP neurons. As such, while we understand the importance of obtaining this information, we do not find it to be within the scope of this manuscript.

Reviewer #3 (Remarks to the Author):

My concerns have been adequately addressed by the authors, and I have no additional comments. I appreciate the authors for their efforts in resolving my concerns.

REVIEWERS' COMMENTS

Reviewer #2 (Remarks to the Author):

As the reviewer pointed out from the beginning, simply showing that the target gene is elevated with loss of miR-33 merely shows an association and does not tell us if it is truly causative.

In addition, although the author states that it is necessary to create many genetically modified mice to show causality, simply expressing Cre-inducible siRNA in the AgRP-Cre mice used by the author will show which target genes are most important.

RESPONSE TO The REVIEWERS

We would like to thank the reviewers for her/his positive comments about our study, and we are very grateful for their input during the revision process. Their suggestions helped to strengthen our manuscript.

Reviewer #2 (Remarks to the Author):

As the reviewer pointed out from the beginning, simply showing that the target gene is elevated with loss of miR-33 merely shows an association and does not tell us if it is truly causative. In addition, although the author states that it is necessary to create many genetically modified mice to show causality, simply expressing Cre-inducible siRNA in the AgRP-Cre mice used by the author will show which target genes are most important.

We have acknowledged in the manuscript that the approaches utilized in this manuscript do not allow us to directly demonstrate which targets of miR-33 are responsible for the effects we observe. The factors that we have chosen to discuss as possibly being involved in mediating these effects were chosen because they have previously been demonstrated to be involved in mediating the activity of AgRP neurons. As such, it would be expected that siRNA of these targets would repress the activity of these neurons and could influence the ability of miR-33 to regulate AgRP neuronal activity and regulation of feeding behavior. This would however still not demonstrate that the effects of miR-33 were directly mediated through targeting of this factor as the activity of these neurons would be impaired generally. In order to truly address this, one would need to impede the ability of miR-33 to target individual factors without impacting its function generally. While a cre-mediated target site blocker would in theory allow this, our experience with these constructs has shown them to be both ineffective and non-specific. As such, a genetic model with specific disruption of the miR-33 binding sequence in a specific target would be the only way to impede regulation by miR-33 without effecting the overall function of the target. Additionally, because miRNAs are known to mediate their effects by targeting multiple different mRNA targets that are often in the same or related pathways, a result showing that siRNA knockdown or targeted disruption of miR-33 binding of a specific factor was not sufficient to prevent the ability of miR-33 to regulate neuronal activity would not necessarily mean that this target was not involved in mediating the effects, but simply that it alone is not solely responsible. While we agree that it would be informative to know exactly what targets are involved in mediating these effects, there are serious challenges to addressing this in a meaningful way and we feel that it is beyond the scope of this manuscript.